# FEDERATED ORTHOGONAL TRAINING: MITIGATING GLOBAL CATASTROPHIC FORGETTING IN CONTINUAL FEDERATED LEARNING

**Yavuz Faruk Bakman** [*]
University of Southern California
ybakman@usc.edu

**Duygu Nur Yaldiz** [*]
University of Southern California
yaldiz@usc.edu

**Yahya H. Ezzeldin**
University of Southern California
yessa@usc.edu

**Salman Avestimehr**
University of Southern California
avestime@usc.edu

## ABSTRACT

Federated Learning (FL) has gained significant attraction due to its ability to enable privacy-preserving training over decentralized data. Current literature in FL mostly focuses on single-task learning. However, over time, new tasks may appear in the clients and the global model should learn these tasks without forgetting previous tasks. This real-world scenario is known as Continual Federated Learning (CFL). The main challenge of CFL is *Global Catastrophic Forgetting*, which corresponds to the fact that when the global model is trained on new tasks, its performance on old tasks decreases. There have been a few recent works on CFL to propose methods that aim to address the global catastrophic forgetting problem. However, these works either have unrealistic assumptions on the availability of past data samples or violate the privacy principles of FL. We propose a novel method, Federated Orthogonal Training (FOT), to overcome these drawbacks and address the global catastrophic forgetting in CFL. Our algorithm extracts the global input subspace of each layer for old tasks and modifies the aggregated updates of new tasks such that they are orthogonal to the global principal subspace of old tasks for each layer. This decreases the interference between tasks, which is the main cause for forgetting. We empirically show that FOT outperforms state-of-the-art continual learning methods in the CFL setting, achieving an average accuracy gain of up to 15% with 27% lower forgetting while only incurring a minimal computation and communication cost. Code can be found here.

## 1 INTRODUCTION

Federated learning (FL) is a decentralized training solution born from the need of keeping the local data of clients private to train a global model (McMahan et al., 2017). Most of the FL works focus on the global learning of a single task (Hard et al., 2018; Yang et al., 2021; Augenstein et al., 2020). However, in real life, new tasks might arrive to the clients over time while previous data disappear due to storage limitations. For instance, assume a malware classifier is trained over multiple FL clients. The emergence of new malware families (new tasks) is inevitable, making the update of the classifier a necessity. Another real-life scenario can be the emergence of new viruses in some clients due to epidemics. The global model also has to learn to classify these new viruses (new tasks) (Yoon et al., 2021). In both of these scenarios, the model should not forget its prior knowledge.

Continual Learning (CL) addresses this issue in centralized machine learning (ML), the problem of learning sequentially arrived tasks without forgetting (Kirkpatrick et al., 2017). Learning a global model while new tasks appear in the clients in an online manner is a problem of Continual Federated Learning (CFL) (Ma et al., 2022). An ideal CFL algorithm solving global catastrophic forgetting

---

[*]Equal Contribution

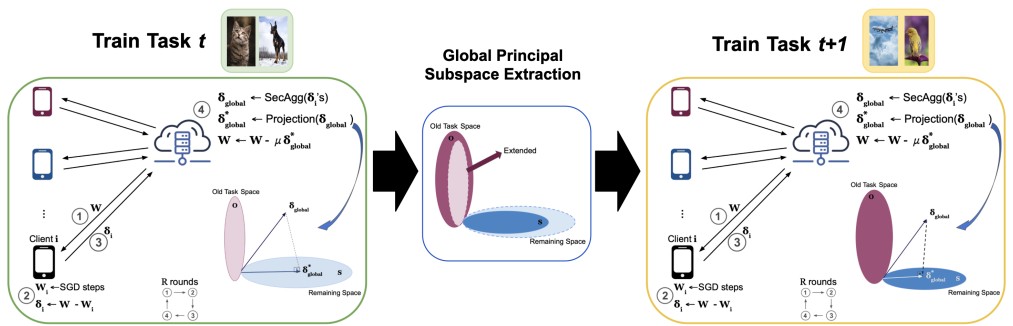

Figure 1: Overview of Federated Orthogonal Training (FOT). Clients do regular SGD training. The server projects the aggregated updates into the orthogonal subspace of previous tasks' activation subspace. At the end of each task, one communication round is reserved to extract global principal activation subspace for each layer.

should not require storage of old task data and an unrealistic amount of computation on the client side because they are already limited in the edge devices. Moreover, it should be compatible with secure aggregation (Bonawitz et al., 2017). The latest research has shown that local data can be extracted from the local updates if there is no protection such as secure aggregation (Boenisch et al., 2021; Wang et al., 2022; Fowl et al., 2022). Lastly, it should be resistant to data heterogeneity across clients which is mostly the case in real-world scenarios (Wang et al., 2021).

**Continual Federated Learning Challenges:** The goal of centralized continual learning algorithms is to prevent the disruption of knowledge acquired from previous tasks while learning new tasks. When the network capacity is fixed, this is accomplished by transferring the information of old tasks to the current learning process and training the model accordingly. Carrying global information of old tasks is a non-trivial problem in Continual Federated Learning due to decentralization. One possible solution is that the clients can maintain old tasks' information according to their local data. However, maintaining local information on the client side requires extra storage. Considering edge device limitations, this approach is not feasible in real life. Moreover, even if it is pursued, mitigation of global forgetting is not guaranteed because clients do not have global information of old tasks. Clients update the model to prevent forgetting according to their local data. However, aggregation of these local updates might not result in preventing global forgetting. This becomes a much major problem in non-IID settings. This effect is visualized in Figure 2 and demonstrated in Appendix A.

**Our Contributions:** In this work, we propose a CFL framework named Federated Orthogonal Training (FOT) to address the *Global Catastrophic Forgetting* problem. Our framework (visualized in Figure 1) modifies the global updates of new tasks so that they are orthogonal to previous tasks' activation principal subspace. It decreases the interference between tasks therefore learning new tasks does not disrupt model performance on old tasks. Within FOT, we introduce a novel aggregation method, named FedProject, which guarantees the orthogonality in a global manner. FedProject requires the global principal subspace information of old tasks for each layer on the server. Therefore, FOT has Global Principal Subspace Extraction (GPSE) to obtain that information. Our method protects the privacy of the clients without any requirement of trusted third parties or representative data in the server. Also, clients do not store anything and there is no extra computation on the local training. Furthermore, our method is robust under data heterogeneity. We perform an extensive comparison of our method with state-of-the-art baselines. Our method outperforms all other methods with a significant margin.

## 2    RELATED WORK

Aiming to train a global model in federated learning for multiple tasks with online data arrivals, the primary challenge is mitigating forgetting while preserving privacy (Ma et al., 2022). Limited prior efforts address this challenge. One approach involves clients storing and sharing perturbed subsets of previous task samples (Dong et al., 2022), but this strains storage and privacy. Knowledge distillation, another solution, relies on a server's access to task-specific datasets (Ma et al., 2022), which may not be practical in certain scenarios. Recent proposals (Babakniya et al., 2023; Qi et al., 2023; Zhang et al., 2023) suggest using generative models to generate synthetic samples to prevent forget-

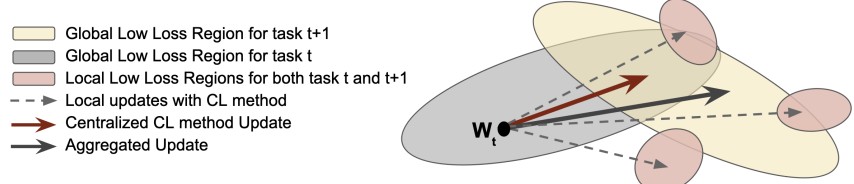

Figure 2: Dashed lines show each client maintains its local information of old tasks and apply a continual learning method locally. The local updates try to reach the loss region where old and new task loss is low locally (red regions). However, some portion of local low-loss regions might not overlap with the global low-loss regions of both tasks (intersection of gray and yellow regions). Therefore, aggregated update (shown as solid black arrow) may converge to the point where old task loss is high (outside of the gray region)

ting, but these add significant computational and communication overhead. Another method (Halbe et al., 2023) employs foundation models and prompt tuning but requires individual client updates and an available foundation model at the server, increasing costs substantially. Our approach, in contrast, combats forgetting without sharing data samples or assuming server-side datasets. It incurs minimal additional computation and communication costs. While related to GPM (Saha et al., 2021) in centralized continual learning, our adaptation introduces a privacy-preserving framework to extract global principal subspaces at each layer, ensuring orthogonality with a novel aggregation method. We also account for data distribution heterogeneity across clients. Other works leverage layer orthogonality in various contexts, such as one-shot federated learning (Su et al., 2023) and unlearning (Li et al., 2023), but differ in their approaches and goals. An extended version of the related work can be found in Appendix B.

## 3 PROBLEM FORMULATION

In a traditional federated learning setup, there is a single task $\mathcal{T}$ and there are $C$ clients to train a global model with parameters $\mathbf{W}$ in $R$ communication rounds. At each round, clients receive the global model from the server and perform local training on their local private data. Then clients send the updated models to the server and the server aggregates the locally updated models (McMahan et al., 2017). The optimization objective is $\min_{\mathbf{W}} \sum_{i=1}^{C} \frac{n_i}{n_{total}} \mathcal{L}(\mathcal{D}_i; \mathbf{W})$ where $\mathcal{D}_i$ is the local private dataset of client $i$, $n_i$ is the number of samples in $\mathcal{D}_i$ and $n_{total} = \sum_i n_i$ is the total number of samples across clients.

In Continual Federated Learning (CFL), there are $K$ tasks ($\{\mathcal{T}_1, \mathcal{T}_2, \cdots, \mathcal{T}_K\}$) sequentially arriving at the $C$ clients. Each client $i$, has a labelled private dataset for each task $\mathcal{T}_t$ denoted as $\mathcal{D}_{t,i} = \{(\mathbf{x}_{t,i}^1, y_{t,i}^1), (\mathbf{x}_{t,i}^2, y_{t,i}^2), ..., (\mathbf{x}_{t,i}^{n_{t,i}}, y_{t,i}^{n_{t,i}})\}$, where $(\mathbf{x}_t^i, y_t^i)$ are the feature and label pair of the $i$-th data sample for task $t$, and $n_{t,i}$ is the number of data samples on client $i$ for task $\mathcal{T}_t$. The aim is to train $\mathbf{W}$ in a federated manner to learn the sequentially arrived $K$ tasks. More formally, in CFL we aim to solve the following optimization problem

$$\min_{\mathbf{W}} \sum_{t=1}^{K} \sum_{i=1}^{C} \frac{n_{t,i}}{\sum_{i=1}^{C} n_{t,i}} \mathcal{L}_t(\mathcal{D}_{t,i}; \mathbf{W}), \tag{1}$$

The objective aims to minimize the loss of all tasks among all samples, where each task is evaluated separately (i.e. task-incremental learning). We consider that each task $\mathcal{T}_i$, appears for $R_i$ communication rounds, during which only data from $\mathcal{T}_i$ is available at the clients.

The challenge in solving (1) stems from the nature of continual learning: $\mathcal{D}_{t,i}$ is available only during the learning of $\mathcal{T}_t$. Neither clients nor the server can access the old tasks data. Furthermore, we consider that there is no representative data for any of the tasks at the server. Due to the nonavailability of old tasks, the model forgets what it learns from previous tasks. Our central goal is to solve this *global catastrophic forgetting* phenomenon while still preserving the privacy of each of the clients' local datasets. To study this problem, following the existing literature (van de Ven & Tolias, 2019; Saha et al., 2021; Augenstein et al., 2020; Dong et al., 2022; Kirkpatrick et al., 2017), we focus on the extreme forgetting case where the tasks are clearly separated, i.e., there is no intersection between rounds where tasks $\mathcal{T}_i$ and $\mathcal{T}_j$, $\forall i \neq j$.

## 4 PROPOSED SOLUTION

Let $\mathbf{W}$ be composed of $L$ layers, $\mathbf{W} = \{\mathbf{W}^\ell\}_{l=1}^L$ where $\mathbf{W}^\ell$ is the parameters of layer $\ell$ and $\mathbf{X}_t^\ell$ be the input matrix for layer $\ell$ at task $\mathcal{T}_t$. Each column corresponds to one input for the layer and these inputs are distributed among the $C$ clients $\mathbf{X}_t^\ell = [\mathbf{x}_{t,1}^{1,\ell}, \mathbf{x}_{t,1}^{2,\ell}, .., \mathbf{x}_{t,1}^{n_{t,1},\ell}, \mathbf{x}_{t,2}^{1,\ell}, ..., \mathbf{x}_{t,2}^{n_{t,2},\ell}, ...., \mathbf{x}_{t,C}^{n_{t,C},\ell}]$. Notice that for the first layer, the columns of $\mathbf{X}_t^1$ are the samples distributed among clients. Now, applying the model to all inputs to layer $\ell$ across the clients can be written as:

$$\mathbf{H}_t^\ell = \mathbf{W}^\ell \mathbf{X}_t^\ell \tag{2}$$

where $\mathbf{H}_t^\ell$ is the output of layer $\ell$ at task $\mathcal{T}_t$ before applying non-linearity. Our aim is to not change $\mathbf{H}_t^\ell$ as much as possible for each layer while training new tasks ($\{\mathcal{T}_{t+1}, \mathcal{T}_{t+2}, .., \mathcal{T}_K\}$).

### 4.1 FEDERATED ORTHOGONAL TRAINING (FOT)

To present the central idea of FOT, let us focus on the training of the first two tasks $\mathcal{T}_1$ and $\mathcal{T}_2$. With no previous tasks to care about, the training of $\mathcal{T}_1$ follows the vanilla FedAvg (McMahan et al., 2017). Let $\mathbf{W}_t = \{\mathbf{W}_t^\ell\}_{\ell=1}^L$ be the global model parameters for layer $\ell$ after concluding the training of $\mathcal{T}_t$. While training $\mathcal{T}_2$, $\mathbf{W}_1^\ell$ is updated to $\mathbf{W}_1^\ell + \Delta\mathbf{W}^\ell$, $\forall\ell \in [1:L]$. Now consider the layer outputs at layer $\ell$ when applying the new global model parameters on data from $\mathcal{T}_1$:

$$\mathbf{H}_1^{\ell\,\star} = (\mathbf{W}_1^\ell + \Delta\mathbf{W}^\ell)\mathbf{X}_1^\ell, \tag{3}$$

where $\mathbf{X}_1^\ell$ is the inputs to layer $\ell$ assuming samples from task $\mathcal{T}_1$. Our goal for task $\mathcal{T}_1$ is to achieve that $\mathbf{H}_1^{\ell\,\star} = \mathbf{H}_1^\ell$, $\forall\ell \in [1:L]$, such that the layer outputs and therefore the model mapping on data from task $\mathcal{T}_1$ is unchanged due to subsequent tasks. From (3), we can see that this is achieved when $\Delta\mathbf{W}^\ell\mathbf{X}_1^\ell = 0$ as in this case there is no interference from other tasks affecting $\mathcal{T}_1$. More generally, we would like to remove interference to task $T_t$ by any of its subsequent tasks $T_q$, with $q > t$, i.e., for any task $t$, we have that:

$$\Delta\mathbf{W}_q^\ell\mathbf{X}_t^\ell = \mathbf{0}, \quad \forall q \in [t+1:T], \ \forall\ell \in [1:L]. \tag{4}$$

Note, however, that making $\Delta\mathbf{W}_q^\ell\mathbf{X}_1^\ell = 0$ requires the row space of $\Delta W_q^\ell$ to be orthogonal to the column space of $\mathbf{X}_t^\ell$. When there are enough training samples for task $\mathcal{T}_t$ the column space of $\mathbf{X}_1^\ell$ is full rank, which enforces $\Delta\mathbf{W}_q^\ell = 0$, effectively blocking learning for any subsequent tasks after $\mathcal{T}_t$ (no training). Therefore, instead of making $\Delta\mathbf{W}^\ell\mathbf{X}_1^\ell = \mathbf{0}$, we will enforce a relaxed condition by requiring $\Delta\mathbf{W}^\ell\mathbf{P}_{X,1}^\ell = 0$, where $\mathbf{P}_{X,1}^\ell$ is the low-rank Global Principal Subspace (GPS) of $\mathbf{X}_1^\ell$. This subspace includes most of the $\mathbf{X}_1^\ell$. We use the term "Global" since this subspace is extracted from the whole data distributed among clients instead of the local data of individual clients.

#### 4.1.1 FEDERATED PROJECTED AVERAGE (FEDPROJECT)

Extracting global principal subspace for each layer in a privacy-preserving manner is a challenging problem which we address next in Section 4.1.2. For now, let us assume that we have $\mathcal{O}_1^\ell$ which denotes the set of orthogonal vectors covering the GPS of layer $\ell$ after training for $\mathcal{T}_1$. While training $\mathcal{T}_2$, aggregated updates are projected onto the orthogonal subspace of $\mathcal{O}_1^\ell$ for each layer $\ell$ as follows:

$$\delta_{global}^\ell \leftarrow \frac{1}{C}\sum_{i=1}^C \delta_i^\ell \tag{5}$$

$$\delta_{global}^{\ell*} \leftarrow \delta_{global}^\ell - \mathbf{O}_1^\ell \mathbf{O}_1^{\ell\,T} \delta_{global}^\ell \tag{6}$$

$$\mathbf{W}^\ell \leftarrow \mathbf{W}^\ell - \mu\delta_{global}^{\ell*} \tag{7}$$

where $\delta_i^\ell$ denotes local update of client $i$ for layer $\ell$ and $\mathbf{O}_1^\ell$ is a matrix whose columns correspond to basis vectors of $\mathcal{O}_1^\ell$. (5) is done with secure aggregation (Bonawitz et al., 2017). By using the projection in (6), we remove the component of the updates that can disrupt the previous task knowledge of the model, and make the model learn the new task in a subspace that is not effectively used in previous tasks.

The local training of clients is not modified. They perform regular training on their local data. After $\mathcal{T}_2$ is finished, $\mathcal{O}_1^\ell$ is expanded such that it also covers the global principal subspace of $X_2^\ell$ for each layer $\ell$ and it is denoted as $\mathcal{O}_2^\ell$. For upcoming tasks, aggregated updates are projected into to the orthogonal subspace of previous tasks' GPS for each layer. After the training of each task is finished, the orthogonal set is expanded again. The orthogonal set expansion and Extraction of Global Principal Subspace are covered in Section 4.1.2. Overall FOT is summarized in the Appendix D and visualized in Figure 1.

**Remark 1.** Note that FedProject is guaranteed to converge for each task under the assumptions stated in (Li et al., 2020b). This is due to two key aspects. First, the projection step in FedProject projects the update onto a set defined by $O_t$. As a result, applying FedProject with gradient updates is equivalent to applying ProximalSGD (Cohen et al., 2017) for a constrained convex optimization problem, which is guaranteed to converge if the projection set is convex (we prove that the projection set in FedProject is convex in Appendix C). Secondly, as shown in (Li et al., 2020b), even when clients perform a number of local steps locally before aggregation, if we consider a small enough learning rate, the local updates can be approximated by a single gradient step with a larger learning rate. This coupled with our equivalence to ProximalSGD, guarantees the convergence even when projection is performed after a number of local epochs.

### 4.1.2 GLOBAL PRINCIPAL SUBSPACE EXTRACTION (GPSE)

After the training for task $\mathcal{T}_t$ is done, the server and clients go through an additional communication round in order to extract the global principal subspace information for each layer in the model. In this additional round, denoted as the GPSE round, the server broadcasts to all clients the global model parameters $\mathbf{W}_t$ and orthogonal vector set $\mathcal{O}_{t-1} = \{\mathbf{O}_{t-1}^1, \cdots, \mathbf{O}_{t-1}^L\}$ that covers the global principal subspace (GPS) of each model layer to the clients. Each client $i$ applies the global model $\mathbf{W}_t$ on its local dataset, then for each layer $\ell$, it projects the layer inputs onto the subspace orthogonal to $\mathcal{O}_{t-1}^\ell$ as follows:

$$\mathbf{X}_{t,i}^\ell{}^* \leftarrow \mathbf{X}_{t,i}^\ell - \mathbf{O}_{t-1}^\ell \mathbf{O}_{t-1}^{\ell}{}^T \mathbf{X}_{t,i}^\ell \tag{8}$$

Where $\mathbf{X}_{t,i}^\ell$ is the input matrix of layer $\ell$ of client $i$ at task $t$. With (8), $\mathbf{X}_{t,i}^\ell{}^*$ across the different clients span a subspace that was not explored by the tasks up to $\mathcal{T}_{t-1}$.

The goal now is to estimate the global principal subspace of $\left\{\mathbf{X}_{t,i}^\ell{}^*\right\}_{i=1}^C$ for each layer in order to add it to $\mathbf{O}_{t-1}^\ell$. To do this each, we will invoke results from randomized SVD to estimate the global principal subspace without violating privacy. First, each client $i$ multiplies projected input vectors with a random row vector and sums them up:

$$\mathbf{A}_{t,i}^\ell \leftarrow \sum_{j=1}^{n_{t,i}} \mathbf{x}_{t,i}^{j,\ell}{}^* \mathbf{g}_j^{\ell}{}^T \tag{9}$$

where $\mathbf{g}_j^\ell$ is a standard normal random vector of length $s^\ell$ (sampling dimension) sampled from $\mathcal{N}(\mathbf{0}, \mathbf{I})$. Note that $\mathbf{A}_{t,i}^\ell$ is a matrix of size $d^\ell \times s^\ell$ where $d^\ell := dim(x^\ell)$. Each client $i$ sends $\{A_{t,i}^\ell\}_{\ell=1}^L$ to the server, which then sums all $\mathbf{A}_{t,i}^\ell$ matrices for each layer $\ell$:

$$\mathbf{A}_t^\ell \leftarrow \sum_{i=1}^C \mathbf{A}_{t,i}^\ell \tag{10}$$

The summation in (10) is done with Secure Aggregation (Bonawitz et al., 2017), thus ensuring that the server only has access to the aggregate $\mathbf{A}_t^\ell$ (See the discussion in Section 4.2). Note that the resulting $\mathbf{A}_t^\ell$ can be written as:

$$\mathbf{A}_t^\ell = \sum_{i=1}^C \sum_{j=1}^{n_{t,i}} \mathbf{x}_{t,i}^{j,\ell}{}^* \mathbf{g}_{j,i}^{\ell}{}^T = \mathbf{X}_t^{\ell*} \times \mathbf{G}^\ell \tag{11}$$

where $\mathbf{X}_t^{\ell*} = [\mathbf{X}_{t,1}^\ell{}^*, \cdots, \mathbf{X}_{t,C}^\ell{}^*]$ is the projected $\mathbf{X}_t^\ell$ to the subspace orthogonal to $\mathbf{O}_{t-1}^\ell$ and $\mathbf{G}^\ell$ is a standard normal Gaussian matrix with $N_t \times s^\ell$ dimension; $N_t$ represents the total number of samples among all clients at task $t$ (i.e. $N_t = \sum_{i=1}^C n_{t,i}$).

With (11) in mind, we can now use an important result in randomized linear algebra, which we state informally in the following theorem for brevity.

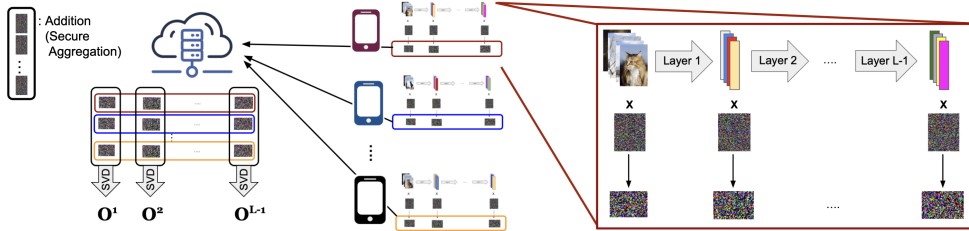

Figure 3: Summary of GPSE. At the end of each task, clients multiply layer inputs with a random gaussian matrix. These matrices are securely summed in the server. SVD is applied to the resulting matrix to get the principal subspace.

**Theorem 1.** *Papadimitriou et al. (1998) (Informally Stated): Let $\mathbf{A}$ be an $m \times n$ matrix and $\mathbf{G}$ be a random Gaussian matrix of size $n \times p$ where $p \geq k$. Let $\mathbf{Y} = \mathbf{AG}$. Then, for sufficiently large $n$, the principal column space of $\mathbf{Y}$ recovers the low-rank column space of $\mathbf{A}$ up to rank $k$ with a negligible error. The error decreases as $p$ increases.*

From Theorem 1, the server can obtain the principal subspace of $\mathbf{X}_t^{\ell*}$ approximately by using $\mathbf{A}_t^\ell$. To find the subspace where most of the inputs lie on, the server applies SVD on $\mathbf{A}_t^\ell$ for each layer $\ell$: $U_t^\ell, \Sigma, V = \texttt{SVD}(\mathbf{A}_t^\ell)$. Then server finds the sufficient rank for each layer $\ell$:

$$\min \quad r^\ell \in \mathbb{N} \quad \text{s.t.} \quad \frac{||\mathbf{PP}^T\mathbf{A}_t^\ell||_F}{||\mathbf{A}_t^\ell||_F} + \frac{||\mathbf{X}_t^{\ell*}||_F}{||\mathbf{X}_t^\ell||_F} > th_t^\ell \tag{12}$$

where $\mathbf{P} \leftarrow \mathbf{U}_t^\ell[0 : r^\ell]$ and $th_t^\ell$ is a threshold value. $\frac{||\mathbf{X}_t^{\ell*}||_F}{||\mathbf{X}_t^\ell||_F}$ measures how much of the input matrix is already covered by $\mathbf{O}_{t-1}^\ell$ and $\frac{||\mathbf{PP}^T\mathbf{A}_t^\ell||_F}{||\mathbf{A}_t^\ell||_F}$ measure how much of the remaining input for each layer is approximated. Therefore, the threshold value determines how much portion of the input in total for each layer is covered by the subspace. How to find $\frac{||\mathbf{X}_t^{\ell*}||_F}{||\mathbf{X}_t^\ell||_F}$ is explained in the Appendix E. After determining the minimum sufficient rank $r^\ell$, the server extends the orthogonal vector set for each layer as follows $\mathcal{O}_t^\ell \leftarrow \mathcal{O}_{t-1}^\ell \cup \mathbf{U}_t^\ell[:, 1 : r^\ell]$. Finally, $\mathcal{O}_t^\ell \leftarrow \texttt{GramSchmidt}(\mathcal{O}_t^\ell)$ is applied in order to keep all the vectors orthogonal to each other since the vectors in $\mathcal{O}_{t-1}^\ell$ and $\mathbf{U}_t^\ell[:, 1 : r^\ell]$ are not necessarily orthogonal to each other. The different steps of GPSE are summarized in Appendix D and visualized in Figure 3.

## 4.2 Discussion of Federated Orthogonal Training

**Privacy:** Our algorithm is privacy-preserving. During the FL training rounds, the clients perform regular training on their local devices and send only the updates to the server. Our algorithm allows the use of secure aggregation (Bonawitz et al., 2017) at these rounds because *FedProject* is applied over aggregated updates. More importantly, GPSE rounds are also compatible with secure aggregation (Bonawitz et al., 2017). Each client sends a random Gaussian matrix for each layer and the server sums these random Gaussian matrices as (10) with secure aggregation. Secure aggregation is a well-studied primitive in federated learning that gives information-theoretic guarantees that the server learns nothing about individual $\mathbf{A}_i$ except their sum from their encoded values $\mathbf{Z}_i$, i.e., $I(\{\mathbf{A}_i\}_{i=1}^N; \{\mathbf{Z}_i\}_{i=1}^N | \sum_{i=1}^N \mathbf{A}_i) = 0$. Recently it has been shown in Elkordy et al. (2022) that the leakage through the sum is limited and is upper-bounded by a decreasing function in the number of clients $N$. These guarantees through secure aggregation ensure the privacy of the proposed approach towards leakage about any individual clients. Furthermore, from the server's perspective, GPSE results in a distributed application of the Johnson-Lindenstrauss (JL) transform, which has been shown to provide differential privacy guarantees (Blocki et al., 2012).

**Communication Cost:** When performing training rounds for task $t$ with FOT, there is no additional communication overhead since only local model updates are aggregated at the server using secure aggregation, similar to FedAvg. The server then projects the aggregated update onto the set $\mathcal{O}_t$. When transitioning from task $t$ to task $t + 1$, an additional communication cost is incurred to update the orthogonal set from $\mathcal{O}_t$ to $\mathcal{O}_{t+1}$, by performing an additional secure aggregation round in order to

compute $\mathcal{O}_{t+1}$ (See Section 4.1.2). Since, the size of $\mathbf{A}_t^\ell$ is equal to $d^\ell \times s^\ell$, then the communication overhead in the GPSE step is $O(d \times s^{\max})$, where $d = \sum_\ell d^\ell$ and $s^{\max} = \max(s^\ell)$.

**Computation Cost:** The computation cost of the algorithm on the client-side is negligible. During the training of a task, regular training is performed on the clients. At the end of tasks, clients extract internal activations and multiply with them a random row vector as in (9). This operation is not computationally heavy. Assume there are at most $N$ data points per task, then the computation complexity is $O(N \times d \times s^{\max})$, where $d = \sum_\ell d^\ell$ and $s^{\max} = \max(s^\ell)$. This is close to one local epoch forward pass complexity which is $O(N \times d \times d^{\max})$, where $d^{\max} = \max(d^\ell)$. The difference comes from the ratio of $\frac{s^{\max}}{d^{\max}}$ which is a constant in our experiments.

**Data Heterogeneity:** Our algorithm is not negatively affected by data heterogeneity. The server obtains the Global Principal Subspace of layer inputs by applying SVD on $\mathbf{A}_t^\ell$ for $\mathcal{T}_t$. $\mathbf{A}_t^\ell$, the output of (11), is independent of the data distribution because of the commutative property of addition. That means, for any possible data distribution, the extracted global principal subspace information remains the same. This makes our algorithm resilient to data heterogeneity.

## 5 EXPERIMENTS

### 5.1 EXPERIMENTAL SETUP

Here we provide a summary of the experimental setup with additional details on architectures, hyperparameters, and further details delegated to Appendix F.

**Benchmarks:** We evaluate the performance of our algorithm on four different CL benchmarks. Permuted MNIST (Ebrahimi et al., 2020) is a variant of MNIST (Lecun et al., 1998) containing 10 tasks. Each task is a randomly pixel-wise permuted version of the original MNIST inputs. Split-CIFAR100 (Krizhevsky, 2009) is created by randomly dividing 100 classes into 10 tasks of 10 classes. Split Mini-Imagenet (Saha et al., 2021; Chaudhry et al., 2019a) is created by randomly dividing 100 classes into 20 tasks of 5 classes. Lastly, we use a sequence of 5-Datasets (Saha et al., 2021), where each dataset is considered a different task. These datasets are CIFAR-10 (Krizhevsky, 2009), MNIST, SVHN (Netzer et al., 2011), notMNIST (Bulatov, 2011) and Fashion MNIST (Xiao et al., 2017), and all of them contain 10 classes.

**Baselines:** We compare our method with the Federated Learning adaptations of state-of-the-art Continual Learning solutions, which are EWC (Kirkpatrick et al., 2017), ER (Chaudhry et al., 2019b), RGO (Liu & Liu, 2022) and GPM (Saha et al., 2021). As a direct CFL approach, we compare to GLFC (Dong et al., 2022), TARGET (Zhang et al., 2023) and FedCIL (Qi et al., 2023). However, since GLFC and FedCIL are proposed for class-incremental setup, we adapted them to task-incremental setting with slight changes. We explain the details of the adaptations in the Appendix F.2. Lastly, we only use FedAvg (FL) to demonstrate the worst-case baseline.

**Data Distribution:** We distribute each task's data to the clients in both i.i.d. and non-i.i.d manner. To simulate data heterogeneity, we follow the method of (McMahan et al., 2017). The data is sorted according to the labels, then it is divided into equal-sized shards, where the number of shards is twice the number of clients. Each client is randomly given 2 shards in a non-replicative manner. In the end, each client has data from at most two classes.

**Metrics:** We use two metrics to evaluate the performance of the algorithm. First, we measure the average accuracy (ACC) (Lopez-Paz & Ranzato, 2017) of all tasks at the end of the whole CFL process. Second, we use average forgetting (FGT) (Chaudhry et al., 2018) to evaluate global catastrophic forgetting. Let $a_{i,t}$ be the global model accuracy of task $\mathcal{T}_i$ after the training of task $\mathcal{T}_t$, FGT and ACC are defined as:

$$\text{ACC} = \frac{1}{K} \sum_{i=1}^{K} a_{i,K}, \qquad \text{FGT} = \frac{1}{K-1} \sum_{i=1}^{K-1} a_{i,i} - a_{i,K} \tag{13}$$

### 5.2 EXPERIMENTAL RESULTS

Table 1 provides average accuracy and forgetting results and Table 2 compares baselines in terms of storage at the edge and the extra computation on local training. Federated Orthogonal Training, in

Table 1: Performance results of different methods on various datasets. ACC (higher is better) stands for average classification accuracy of each task, while FGT (lower is better) denotes average forgetting as in (13). We run each experiment 3 times and provide mean and standard deviation.

| | Method | PMNIST | | CIFAR100 | | 5-Datasets | | Mini-Imagenet | |
|---|---|---|---|---|---|---|---|---|---|
| | | ACC(%) | FGT(%) | ACC(%) | FGT(%) | ACC(%) | FGT(%) | ACC(%) | FGT(%) |
| IID | FL | 85.68±0.57 | 8.29±0.62 | 63.12±0.48 | 13.57±1.57 | 77.95±0.58 | 13.46±1.26 | 50.43±1.97 | 33.08±1.96 |
| | EWC+FL | 86.74±0.97 | 8.84±1.40 | 63.13±0.65 | 13.48±1.77 | 77.68±0.55 | 13.76±1.74 | 47.00±1.46 | 36.68±1.57 |
| | ER+FL | 88.61±0.50 | 6.62±0.06 | 65.42±0.49 | 11.72±0.39 | 80.01±1.17 | 10.11±2.68 | 55.26±2.95 | 27.80±3.23 |
| | RGO+FL | 89.81±0.20 | 4.84±0.55 | 63.91±0.53 | 14.39±0.54 | 84.86±1.24 | 3.47±0.52 | 51.00±1.88 | 31.76±1.87 |
| | GPM+FL | 88.27±0.75 | 6.88±0.63 | 59.43±0.48 | 18.13±0.59 | 80.28±1.02 | 10.7±0.61 | 50.26±1.90 | 30.04±1.22 |
| | GLFC | 83.76±1.05 | 14.70±1.13 | 65.16±0.48 | 11.74±0.59 | 81.64±0.22 | 10.07±0.61 | 59.26±1.23 | 23.73±1.27 |
| | TARGET | 89.23±0.37 | 3.13 ±0.36 | 65.28±0.58 | 9.89±0.63 | 82.11±0.44 | 6.64±0.57 | 58.45±1.33 | 25.12±1.29 |
| | FedCIL | 87.43±0.45 | 10.01±0.59 | 60.12±1.10 | 17.03±1.34 | 82.43±0.34 | 7.65±0.36 | 55.98±1.67 | 28.90±1.84 |
| | FOT(Ours) | **90.35**±0.06 | **1.75**±0.06 | **71.90**±0.06 | **0.87**±0.10 | **85.21**±0.93 | **1.11**±0.31 | **69.07**±0.73 | **0.19**±0.11 |
| non-IID | FL | 80.06±0.43 | 9.21±0.67 | 62.56±0.17 | 10.49±0.40 | 67.71±7.75 | 21.55±9.73 | 41.00±0.41 | 32.92±1.25 |
| | EWC+FL | 81.14±1.15 | 10.78±0.96 | 62.57±0.47 | 10.41±0.83 | 68.41±3.14 | 21.42±4.19 | 41.09±2.33 | 32.23±1.20 |
| | ER+FL | 82.77±1.56 | 9.32±2.57 | 58.57±1.59 | 14.78±2.18 | 70.68±2.46 | 18.48±3.27 | 49.23±2.09 | 24.40±0.50 |
| | RGO+FL | 79.31±3.48 | 15.72±3.96 | 40.83±2.86 | 31.57±1.61 | 77.92±0.80 | 6.89±1.12 | 40.43±0.23 | 33.24±0.80 |
| | GPM+FL | 72.17±1.22 | 22.6±2.54 | 34.15±2.31 | 34.60±2.87 | 72.21±0.97 | 16.84±2.15 | 29.63±3.12 | 40.73±4.20 |
| | GLFC | 84.06±0.24 | 12.13±0.54 | 59.89±1.15 | 15.36±1.69 | 77.80±0.49 | 11.06±1.15 | 50.41±1.85 | 23.46±2.21 |
| | TARGET | 84.78±0.61 | 3.70±0.74 | 63.87±1.71 | 8.76±1.65 | 78.16±0.43 | 7.65±0.36 | 53.67±1.99 | 26.31±2.12 |
| | FedCIL | 83.23±0.54 | 8.65±0.69 | 56.23±1.93 | 21.32±2.01 | 77.93±0.41 | 8.01±0.31 | 51.12±2.11 | 25.66±2.35 |
| | FOT(Ours) | **85.21**±0.13 | **1.97**±0.22 | **66.31**±0.25 | **0.60**±0.43 | **79.23**±1.02 | **0.65**±0.27 | **62.06**±0.59 | **0.17**±0.27 |

Table 2: Comparison of baselines in terms of storage on the edge devices and extra computation on local training.

| | Storage at Edge | Extra Computation |
|---|---|---|
| FL | ✗ | ✗ |
| EWC+FL | ✓ | ✓ |
| ER+FL | ✓ | ✓ |
| RGO+FL | ✓ | ✓ |
| GPM+FL | ✓ | ✓ |
| GLFC | ✓ | ✓ |
| FedCIL | ✓ | ✓ |
| FOT(Ours) | ✗ | ✗ |

Table 3: Percentage of used subspace at the end of whole training for the average of all layers.

| | Average (IID) | Average (non-IID) |
|---|---|---|
| PMNIST | 62.18% | 61.36% |
| 5-Datasets | 53.97% | 45.64% |
| CIFAR100 | 47.59% | 32.96% |
| Mini-Imagenet | 89.94% | 86.44% |

Table 4: Running time of a GPSE round and 1 Local Epoch on the client side.

| | 1 Local Epoch(s) | GPSE (s) |
|---|---|---|
| PMNIST | 0.067 | 0.024 |
| CIFAR100 | 0.074 | 0.172 |
| Mini-Imagenet | 0.194 | 0.215 |

terms of average accuracy, has considerably better performance than the FL adaptations of state-of-the-art CL methods in continual federated learning setup. Also, FOT outperforms GLFC and FedCIL in all the benchmarks without any client storage and extra computation. Besides, FOT achieves very small forgetting in all datasets in both i.i.d and non-i.i.d settings. These results validate that FOT successfully transfers old tasks' GPS information and does orthogonal training accordingly. The higher accuracy margin in the non-i.i.d setting is mainly because of extraction of the global subspace information which is independent of distribution of the data (11) and FedProject guarantees the orthogonality in a global manner. On the other hand, when we apply CL methods locally, information stored for old tasks does not reflect their global distribution, which is particularly true when data distributions are heterogeneous. This makes the global model perform worse in heterogeneous settings. In addition to data-heterogeneity across clients per task, we explore the effect of task-heterogeneity across clients (i.e each client does not necessarily train on all tasks) in Appendix G.1. More detailed results of Table 1 can be found in Appendix G.2. Lastly, we provide the running time and computation cost of GPSE in Table 4 and 5. GPSE is performed only once at the end of each task. The computation cost is close to one epoch training and the communication cost is less than the model size. Therefore, FOT has a negligible effect on the costs of client side.

### 5.2.1 ANALYZING THE SUBSPACE OF LAYERS

In this section, we analyze how much portion of the subspace at layers is used by FOT at the end of each task (Table 3). It is expected that the remaining space might not be sufficient for the proper learning of new tasks. However, the learning capability is limited by the network capacity as layer subspace is scaled with network size. Besides, even with 20 tasks in Mini-Imagenet, our method achieves superior performance while there is still space for new tasks, which shows that our method

Table 5: Size (in MB) of the model and the matrices used by FOT.

| | Model Size | $\{\mathbf{A}^\ell\}_{\ell=1}^L$ | $\{\mathbf{O}^\ell\}_{\ell=1}^L$ |
|---|---|---|---|
| Resnet-18 | 2.56 | 0.64 | 0.08 |
| Alexnet | 1.42 | 0.24 | 0.048 |

Table 6: Split Mini-Imagenet results of Gaussian Mechanism in GPSE round with ($\epsilon = 5, \delta = 10^{-5}$)-DP and freezing the first layer after first task (FRZ).

| | IID | | non-IID | |
|---|---|---|---|---|
| | ACC(%) | FGT(%) | ACC(%) | FGT(%) |
| DP | 66.77 | 0.31 | 60.81 | 0.13 |
| FRZ | 69.30 | 0.15 | 62.72 | 0.12 |

can scale with high number of tasks. The behavior of subspace expansion is controlled by the threshold value in (12). When the threshold is bigger, there is less forgetting but it also restricts the available space for new tasks. If it is smaller, it will allow the network to learn more task, but by forgetting old tasks more. The threshold value creates a trade-off between the amount of forgetting and the number of tasks to learn. We investigate the effect of threshold value in Appendix G.3.

### 5.2.2 ADDITIONAL MECHANISMS TO ENHANCE PRIVACY

As discussed in Section 4.2, applying our GPSE mechanism does provide differential privacy since it is a distributed application of the Johnson-Lindenstrauss (JL) transform. However concrete privacy guarantees in this case are dependent on the properties of the data matrix $X$. To further formalize our privacy guarantees, in this subsection, we apply the Gaussian mechanism to the output of the GPSE mechanism (See additional details in Appendix H). As shown in Table 6, applying the Gaussian mechanism causes a decline in performance. However, our FOT still outperforms other baselines both in the IID and the non-IID settings. Furthermore, we make additional experiments that we are freezing the first layer of the model after the first task. In that way, input of the first layer (samples) does not attend in GPSE rounds and only the internal layer's input (activations) is processed in GPSE. This modification does not hurt the performance of FOT as shown in Table 6. Different versions of FOT can be employed depending on the application's required privacy.

## 6 LIMITATIONS

Our work is limited to solving the forgetting problem under the assumption of knowing task boundaries in the task incremental setting. The problem of not knowing task boundaries is an ongoing research area in Continual Learning which is known as Task-Free Continual Learning (Aljundi et al., 2019). Typical continual learning and federated continual learning papers (Ma et al., 2022; Chaudhry et al., 2019a; Saha et al., 2021; van de Ven & Tolias, 2019; Qi et al., 2023; Babakniya et al., 2023; Zhang et al., 2023) assume the boundaries are known and are well-separated. We follow the literature yet acknowledge that finding task boundaries is an important problem and can be a topic of further research. Lastly, we investigate task-heterogeneity cases where each client sees different subsets of tasks. These experiments are provided in Appendix G.1. We observe that once we know the task boundaries, task heterogeneity is not a problem for FOT.

## 7 CONCLUSION

In this work, we propose Federated Orthogonal Training to mitigate the *Global Catastrophic Forgetting* problem in Continual Federated Learning, where FL clients receive new tasks over time and the global model performance on old tasks diminishes. Alleviating global forgetting is a challenging problem due to the distributed and privacy-preserving nature of FL. To the best of our knowledge, our approach is the first solution for CFL that mitigates global forgetting without requiring extra storage and major additional computation at edge devices. Moreover, we protect the client's privacy since FOT is compatible with secure aggregation. We evaluate our algorithm on different CFL benchmarks with various network architectures and compare it with the FL adaptations of state-of-the-art CL solutions. Experimental results show that our method mitigates global forgetting while achieving high accuracy performance. We also present that the extra communication and computation costs of our algorithm are negligible.

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

APPENDIX

## A    ANALYZING THE EFFECT OF DECENTRALIZATION ON FORGETTING

In this section, we demonstrate the main challenge of CFL which is carrying global information of old tasks. To demonstrate this, we compare the performance of ER (Chaudhry et al., 2019b) on Split-Cifar100 (Krizhevsky, 2009) in centralized and federated settings. Federated ER stores some samples from past tasks on the client-side while the centralized ER does the storage and training in a centralized manner. In the federated setup, 125 samples are stored in total among all clients. In the centralized setting, 100 samples are stored in total. We compute the average forgetting difference between federated ER over FedAvg and ER over standard SGD. Table 7 shows that there is more improvement in centralized ER than federated ER in terms of average forgetting with less storage. The performance gap between federated and centralized ER is due to the fundamental challenge in CFL. In the federated setting, old tasks' information is only carried locally and aggregated at the server. This may not preserve old tasks' information globally compared to the centralized approach.

Table 7: Performance improvement of ER over baselines

|                               | Centralized | Federated |
| ----------------------------- | ----------- | --------- |
| Total Number of Samples       | 100         | 125       |
| ER Improvement over Baselines | 2.71%       | 0.48%     |

## B    RELATED WORK

**Centralized Continual Learning.** Several methods for continual learning and dealing with catastrophic forgetting have been recently proposed (De Lange et al., 2019). We can categorize these solutions for the centralized settings into three methods: **(1) Rehearsal-based** approaches replay previous tasks' data that is stored in a limited memory (Lopez-Paz & Ranzato, 2017; Chaudhry et al., 2019a; Rolnick et al., 2019; Guo et al., 2020) or that is generated using generative models (Shin et al., 2017) to reduce forgetting of older tasks. However, the application of these solutions in FL may violate federated learning's data storage and privacy requirements. Li & Hoiem (2016) follows an experience replay based method, where the model at the end of the previous task is stored and used as a teacher, i.e knowledge distillation is applied on the features before the classification layer. The application of this method to CFL overperformed by generative based approaches (Qi et al., 2023; Babakniya et al., 2023). (2) **Expansion-based** approaches increase the model capacity to solve the problem of forgetting. The model is a dynamic network that increases in size as tasks arrive (Rusu et al., 2016; Yoon et al., 2018; Sarwar et al., 2020; Yoon et al., 2020). Distributed application of these methods in FL may result in excessively big models, especially in scenarios where client datasets can be highly heterogeneous. (3) **Regularization-based** approaches modify the direction of gradients for each task by adding regularization such that the optimal parameters of the new task are close to the optimal points of previous tasks (Kirkpatrick et al., 2017; Liu & Liu, 2022; Serra et al., 2018; Mallya & Lazebnik, 2018). These approaches (Kirkpatrick et al., 2017; Liu & Liu, 2022; Serra et al., 2018; Mallya & Lazebnik, 2018) require access to information from previously trained tasks. Another approach for regularization was adopted by GPM (Saha et al., 2021) which modifies the model gradients per layer for the sequential tasks such that the updates are orthogonal to the core gradient subspace of previous tasks. When applied distributively in an FL system, GPM is not guaranteed to achieve orthogonalization in the global model as core gradient subspaces might differ across clients. It also requires extra computation and storage on the client side.

**Continual Federated Learning** aims to train a global model in a federated learning system for multiple tasks as data for different tasks arrives at the clients in an online manner. Similar to the centralized continual learning, the main challenge is how to alleviate forgetting of the previous task by the global model, but while maintaining the privacy requirements of federated learning (Ma et al., 2022). A solution to this problem is not trivial because of the privacy constraints of federated

learning (Ma et al., 2022). A limited set of recent works have attempted to solve this non-trivial problem. Dong et al. (2022) proposes a solution where clients store a subset of the samples from previous tasks and share them perturbed with a proxy server, which can be a limiting factor in terms of storage (particularly for edge devices) and can still leak information through multiple views of the perturbed task samples. Ma et al. (2022) follows an approach based on knowledge-distillation in order to alleviate forgetting of the global model. This assumes that the server has a representative dataset for each task to be used for knowledge distillation. The availability of these datasets at the server may not be feasible, particularly in scenarios where FL is used to train models in the absence of representative centralized distributions.

Recently, (Qi et al., 2023; Babakniya et al., 2023; Zhang et al., 2023) propose using generative models to generate synthetic samples to prevent forgetting. However, these approaches increase the computation and communication overhead significantly because they require training additional generative model which is relatively bigger than the classifier model. Besides, the server generates synthetic samples based on the generative model which can cause privacy leakage. (Halbe et al., 2023) uses foundation model and prompt tuning to prevent forgetting but that approach requires individual updates of clients which cannot preserve privacy, and an available foundation model at the server which is not always realistic. It also increases the computation and communication costs a lot because of the size of the foundation models. Lastly, Yoon et al. (2021) addresses a fundamentally distinct problem from ours while categorized under CFL. It is primarily interested in learning individual local tasks, not learning a global model but utilizing the information of other clients. It employs federated learning not to train a global model but to enhance the continual learning process for individual local tasks by leveraging information from other clients.

Our proposed approach for dealing with forgetting in the global model does not require sharing of the data samples with the server or assumes the availability of a dataset subset at the server. Also, our method's additional computation and communication cost is negligible. Perhaps, the closest related work to ours is GPM (Saha et al., 2021) in centralized continual learning, however unlike in the adaptation of GPM to FL, we introduce a privacy-preserving framework to extract the global principal subspaces of each layer and guarantee the orthogonality with a new aggregation method, while also being resilient to the heterogeneity of data distributions across clients. There are also several works using layer's orthogonality in different problems (Li et al., 2023; Su et al., 2023). (Su et al., 2023) is a one shot federated learning work and does not extract the global subspace of the global model. It extracts a local subspace for each client at the client-side and sends it to the server. Also, they do not make a global update; instead, they update each model separately regarding the local subspace of each client. (Li et al., 2023) uses layer's orthogonality idea in unlearning setting. They directly send scaled samples to the server which violates the privacy and scales the communication cost with number of samples.

## C  Convexity of projection set in FedProject

**Throughout this section, we will abuse notation for ease of presentation, by treating all parameter updates $\delta$ in their vectorized form instead of a matrix, i.e., $\delta^\ell$ is a vector of size $d_\ell$, instead of a matrix whose number of elements is equal to $d_\ell$.**

Let $\mathcal{S}$ be the set projected on in (6), i.e. the set where $\delta^*_{global}$ lives in. We want to prove that $\mathcal{S} \subset \mathbb{R}^d$ is a convex set where $d = \sum_\ell d_\ell$ and $d_\ell$ is the parameter-size of layer $\ell$. In other words, we want to show that $\forall \mathbf{g}^{(1)}, \mathbf{g}^{(2)} \in \mathcal{S}$ and $\alpha \in [0, 1]$, we have that $\alpha \mathbf{g}^{(1)} + (1 - \alpha)\mathbf{g}^{(2)} \in \mathcal{S}$.

Note that the set $\mathcal{S} \subset \mathbb{R}^d$ is the cartesian product of the sets $\{\mathcal{S}_\ell\}_{\ell=1}^L$ and as a result, it is convex if the individual sets $\mathcal{S}_\ell$ are all convex.

Thus, we only need to focus on showing that $\mathcal{S}_\ell$ is convex $\forall \ell$. Recall that, by definition, the set $\mathcal{S}_\ell$ is the set of all vectors orthogonal to all the columns of $\mathbf{O}^\ell$, i.e.

$$\mathcal{S}_\ell = \left\{ \mathbf{u} \in \mathbb{R}^{d_\ell} \,\Big|\, \mathbf{O}^{\ell^T} \mathbf{u} = \mathbf{0} \right\}.$$

Now let $\mathbf{g}_\ell^{(1)}, \mathbf{g}_\ell^{(2)} \in \mathcal{S}_\ell$, then we have that

$$\mathbf{O}^{\ell T}\left(\alpha\mathbf{g}_\ell^{(1)} + (1-\alpha)\mathbf{g}_\ell^{(2)}\right) = \alpha\mathbf{O}^{\ell T}\mathbf{g}_\ell^{(1)} + (1-\alpha)\mathbf{O}^{\ell T}\mathbf{g}_\ell^{(2)}$$

$$= \mathbf{0} \qquad \implies \alpha\mathbf{g}_\ell^{(1)} + (1-\alpha)\mathbf{g}_\ell^{(2)} \in \mathcal{S}_\ell,$$

which proves that $\mathcal{S}_\ell$ is convex.

## D    ALGORITHMS

The Federated Orthogonal Training is summarized in Algorithm 1. The Global Principal Subspace Extraction is summarized in Algorithm 2.

---

**Algorithm 1** Federated Orthogonal Training

---

Let $C$ be number of clients, $T$ be number of tasks, $L$ be number of layers.
$\delta^\ell$ represents update for layer $\ell$
$\delta := \{\delta^\ell\}_{\ell=1}^L$
**Input:** Total communication rounds per task $R_t$ and data of the client $i$ for task $t$ is $\mathcal{D}_{t,i}$, where $i \in [1:C]$ and $t \in [1:T]$
**Initialize:** $\mathcal{O}^\ell = \{\}, \forall \ell \in [1:L]$.
$\mathcal{O} \leftarrow \{\mathcal{O}^\ell\}_{\ell=1}^L$
**for** task $t = 1$ **to** $T$ **do**
   **for** round $r = 1$ **to** $R_t$ **do**
      **Server runs:**
         Send the global model $\mathbf{W}$ to clients
      **Each client i runs:**
         Train model with local data $\mathcal{D}_{t,i}$
         Send update $\delta_i$ to Server
      **Server runs:**
         `FedProject`$(\mathbf{W}, \mathcal{O}, \{\delta_i\}_{i=1}^C)$
   **end for**
   // End of a task .. Server extracts new principal subspace
   $\mathcal{O} \leftarrow \text{GPSE}(\mathbf{W}, \mathcal{O}, \{\mathcal{D}_{t,i}\}_{i=1}^C)$
**end for**

**FedProject**$(\mathbf{W}, \mathcal{O}, \{\delta_i\}_{i=1}^C)$:
   $\delta_{global} = \frac{1}{C}\sum_{i=1}^C \delta_i \leftarrow \text{SecAgg}\left(\{\delta_i\}_{i=1}^C\right)$
   **for** layer $\ell = 1$ **to** $L$ **do**
      // Applying (6) ... $\mathbf{O}^\ell$ is the orthonormal basis for $\mathcal{O}^\ell$
      $\delta_{global}^{\ell*} \rightarrow \delta_{global}^\ell - \mathbf{O}_1^\ell \mathbf{O}_1^{\ell T}\delta_{global}^\ell$
   **end for**
   $\mathbf{W} \leftarrow \mathbf{W} - \mu\delta_{global}^*$

---

## E    COMPUTING PROJECTED PORTION OF THE DATA

After each task is finished, clients project their layer inputs as $\mathbf{X}_{t,i}^{\ell}{}^* = \mathbf{X}_t^\ell - \mathbf{O}_{t-1}^\ell \mathbf{O}_{t-1}^{\ell T}\mathbf{X}_{t,i}^\ell$ where $\mathbf{X}_{t,i}^\ell$ is the input matrix for client $i$ and layer $\ell$ at task $t$. After the projection is done, clients send the ratio $\frac{||\mathbf{X}_{t,i}^{\ell}{}^*||_F}{||\mathbf{X}_{t,i}^\ell||_F}$ to the server, which is the information of how much portion of the input is already in $\mathbf{O}_{t-1}^\ell$. In the served-side, these rates are summed up $\sum_{i=1}^C \frac{||\mathbf{X}_{t,i}^{\ell}{}^*||_F}{||\mathbf{X}_{t,i}^\ell||_F}$. This summation gives an approximation of $\frac{||\mathbf{X}_t^{\ell*}||_F}{||\mathbf{X}_t^\ell||_F}$. Note that this summation can be done with secure aggregation.

---

**Algorithm 2** Global Principal Subspace Extraction (GPSE)

---

**Input:** Global model $\mathbf{W}$, orthogonal set $\mathcal{O}$, and set of data of the clients $\{\mathcal{D}_i\}_{i=1}^{C}$, where $C$ is the number of clients
**Output:** Updated orthogonal set $\mathcal{O}$
**Server runs:**
    Send global model $\mathbf{W}$ and $\mathcal{O}$ to the clients
**Each client i runs:**
    $\mathbf{A}_i \leftarrow \texttt{RandomizedActivationCollection}(\mathbf{W}, \mathcal{O}, \mathcal{D}_i)$
    Send $\mathbf{A}_i$ to the server
**Server runs:**
    $\mathcal{O} \leftarrow \texttt{ExpandOrthogonalSet}(\{\mathbf{A}_i\}_{i=1}^{C}, \mathcal{O})$
    **return** $\mathcal{O}$

**RandomizedActivationCollection**$(\mathbf{W}, \mathcal{O}, \mathcal{D})$:
    Feed local data $\mathcal{D}$ to the model
    Store inputs of each layer $\mathbf{X}^\ell$, $\forall \ell \in [1:L]$
    **for** layer $\ell = 1$ **to** $L$ **do**
        $\mathbf{X}_t^{\ell*} \leftarrow \mathbf{X}_t^\ell - \mathbf{O}_{t-1}^\ell \mathbf{O}_{t-1}^{\ell T} \mathbf{X}_t^\ell$    //Projecting input on orthogonal subspace as in (8)
        $\mathbf{G}^\ell \sim \mathcal{N}(0, \mathbf{I})$                // Normally distributed Gaussian matrix
        $\mathbf{A}^\ell \leftarrow \mathbf{X}^{\ell*} \times \mathbf{G}^\ell$           // Applying (11)
    **end for**
    **return** $\{\mathbf{A}^\ell\}_{\ell=1}^{L}$

**ExpandOrthogonalSet**$(\{\mathbf{A}_i\}_{i=1}^{C}, \mathcal{O})$:
    **for** layer $\ell = 1$ **to** $L$ **do**
        $\mathbf{A}^\ell = \sum_{i=1}^{C} \mathbf{A}_i^\ell \leftarrow \texttt{SecAgg}\left(\{\mathbf{A}_i\}_{i=1}^{C}\right)$
        $\mathbf{U}^\ell, \Sigma^\ell, \mathbf{V}^\ell \leftarrow \texttt{SVD}(\mathbf{A}^\ell)$
        $r^\ell \leftarrow \texttt{CalculateRank}(\mathbf{A}^\ell, \mathbf{U}^\ell)$ // Solving the problem in (12) to find the minimum rank $r^\ell$
        $\mathcal{O}^\ell \leftarrow \mathcal{O}^\ell \cup \mathbf{U}^\ell[0:r^\ell]$
        $\mathcal{O}^\ell \leftarrow \texttt{GramSchmidt}(\mathcal{O}^\ell)$
    **end for**
    **return** $\{\mathcal{O}^\ell\}_{\ell=1}^{L}$

---

# F    EXPERIMENTAL DETAILS.

## F.1    DATASETS

Dataset statistics are provided in Table 8.

## F.2    BASELINE DETAILS

**EWC** (Kirkpatrick et al., 2017) is a regularization-based method and defines important weights as the Fisher Information diagonals. We adapt this approach to FL (EWC+FL) so that each client calculates and stores its own fisher information matrix and uses it in the local training process. The server performs regular FedAvg (McMahan et al., 2017) as FL.

**ER** (Chaudhry et al., 2019b) is a memory-based approach, where a certain number of samples from each task is stored in the memory and the loss of that is equally contributed to the new tasks. For the FL adaptation of this method (ER+FL), we let each client store a certain number of data points from each task and do local training with stored samples and new tasks' samples together. The server performs regular FedAvg (McMahan et al., 2017) as FL.

**RGO** (Liu & Liu, 2022) updates the optimizer iteratively to modify gradients and decreases the interference of tasks. We adapt this method to FL (RGO+FL) so that each client calculates the Hessian matrix on its local data and maintains its own local projection matrix. The server performs regular FedAvg (McMahan et al., 2017) as FL.

Table 8: Dataset statistics.

|  | **PMNIST** | **5-Datasets** (CIFAR10, MNIST, SVHN, notMNIST, FMNIST) |
| --- | --- | --- |
| Input Size | $1\times28\times28$ | $3\times32\times32$ |
| # tasks | 10 | 5 |
| # clients | 125 | 150 |
| # training/task | 50000 | 50000, 60000, 73257, 60000, 16853 |
| # test/task | 10000 | 10000, 10000, 26032, 10000, 1873 |
|  | **CIFAR100** | **Mini-Imagenet** |
| Input Size | $3\times32\times32$ | $3\times84\times84$ |
| # tasks | 10 | 20 |
| # clients | 50 | 50 |
| # training/task | 5000 | 2500 |
| # test/task | 1000 | 500 |

**GPM** (Saha et al., 2021) is applied locally for each client, i.e. clients compute orthogonal set locally at the end of each task and apply projection to their local updates in each round. The server performs regular FedAvg (McMahan et al., 2017) as FL.

**GLFC** (Dong et al., 2022) is designed for class-incremental while our setting is task-incremental. GLFC proposes a method consisting of 3 parts to prevent forgetting:

1. They make clients store some old tasks data and during the training of new tasks, old samples are also used with a normalization factor. This is very similar to ER (Rolnick et al., 2019) with normalization difference.

2. Clients store the previous task's model and calculate KL divergence loss between the output of the new model and old model. This output comes from the last layer. As it is a class-continuous setting, the output dimension of old models is less than the new model. Therefore, they add new dimensions to the old task's model by using the one-hot representation of the label. This approach is specific to class-incremental settings because in task-incremental there are different heads for different tasks so the old model does not have an output for new tasks.

3. They require an additional proxy server for collecting perturbed gradients to generate samples of the clients.

In order to fairly compare with GLFC, we drop 2 and 3 because 2 is not proper for task-continual setting and 3 assumes there is an additional server.

### F.3 ARCHITECTURES

We use the same architecture for all of the methods we compare and the network size is fixed throughout the whole CFL process.

**MLP architecture:** For Permute MNIST, we use a 4-layer MLP as in (Kirkpatrick et al., 2017), where the first 3 fully-connected layers' hidden size is 400. The last layer is the classification layer with output size of 10. We use ReLU as non-linearity and dropout of 0.2 after the first layer and 0.5 for the next 2 layers.

**AlexNet-like architecture:** For Split-CIFAR100 dataset, we use an AlexNet-like architecture similar to GPM (Saha et al., 2021). There are 3 convolutional layer in the network with 64, 128 and 256 filters, and with 4×4, 3×3, and 2×2 kernel sizes, respectively. ReLU is used as non-linearity and 2x2 Max-Pooling is applied after each Conv layer. Conv layers are followed by two fully-connected layers with hidden size of 2048. Dropout of 0.2 is applied for the first 2 layers, and with 0.5 for the rest of the network. The model is multi-headed, where a separate classification layer for each task exists at the end of the network. The output size is 10 for each of the heads.

**Reduced ResNet18 architecture:** For 5-Datasets and Split Mini-Imagenet, we use a reduced ResNet18 architecture similar to the one in (Lopez-Paz & Ranzato, 2017). We subtitute the 4 × 4 Average-Pooling before classifier layer with 2 × 2 Average-Pooling. For Split Mini-Imagenet, we use convolution with stride 2 in the first layer. The model is multi-headed, where a separate classification layer for each task exists at the end of the network. The output size is 5 (Split Mini-Imagenet) and 10 (5-Datasets) for each of the heads.

## F.4 HYPERPARAMETERS

Note that we obtained the optimal hyperparameters presented in our work by performing grid search.

**Local Training Details:** In all of the experiments, local number of epochs is 1 with SGD optimizer and learning-rate=0.01. Batch size is set to 64 for Permute MNIST, and 16 for Split-CIFAR100, 5-Datasets and Split Mini-Imagenet.

**Number of Communication Rounds:** For Permute MNIST, we run vanilla FL for 1100 rounds in total, where first task is run for 200 and the others for 100 rounds. All of the other methods' total communication round is set to 2000, divided equally to the tasks. For 5-Datasets, again each method is run for the same round configuration. Number of rounds are 1000, 100, 200, 100, 100 in the order of 5-Datasets tasks, making in total 1500 rounds. For Split-CIFAR100, total number of round is the same for each method with being 10000 and shared equally among 10 tasks. For Split Mini-Imagent, each task is trained for 250 rounds, making in total of 5000 rounds, and this is set the same for each method.

**Number of Clients:** For Permute-MNIST dataset, we set the number of total clients to 125 and select 64 of them randomly. In this setup each client has 480 data points per task. For 5-Datasets, we consider 150 clients in total and randomly select 64 of them. Local number of data points per client is as follows in the task order: 333, 400, 488, 400, and 112. For Split-CIFAR100 and Split Mini-Imagenet, there are 50 clients in total and the selection rate in in round is 0.5. Local number of data points per client per task is 100 and 50 for Split-CIFAR100 and Split Mini-Imagenet, respectively.

**Method Specific Hyperparameters:**

- **EWC+FL**
  - regularization coefficient: 40 (PMNIST), 20 (CIFAR100), 1 (5-Datasets), 0.01 (Mini-Imagenet)

- **ER+FL**
  - total number of stored data among all clients: 125 (PMNIST), 50 (CIFAR100 and Mini-Imagenet), 150 (5-Datasets)

- **RGO+FL**
  - percentage of clients (randomly selected) that store projection matrix per task: 40% (for all datasets)

- **GLFC**
  - total number of stored data among all clients: 125 (PMNIST), 50 (CIFAR100 and Mini-Imagenet), 150 (5-Datasets)

- **FOT**
  - sampling dimension $s^\ell$ is the same as $d^\ell$ for MNSIT and $5d^\ell$ for 5-Datasets, CIFAR100 and Mini-Imagenet (9) (Li et al., 2020a)

**Threshold values for Federated Orthogonal Training:**

Threshold values for Federated Orthogonal Training (see (12)) are provided in Table 9. Threshold value is the same for each layer of the network. We do not tune the threshold value for each task, we solely set the threshold value for the first task and increment it evenly after each task is finished. The initial threshold values are 0.94, 0.95, 0.93 and 0.90 for PMNIST, 5-Datasets, Split-CIFAR100 and Mini-Imagenet, respectively. The increment value for 5-Datasets, Split-CIFAR100 and Mini-Imagenet is 0.001 while it is zero for Permute MNIST. Note that there are 5 tasks in 5-Datasets, while Permute MNIST and Split-CIFAR10 have 10 and Mini-Imagenet has 20.

Table 9: Threshold values for Federated Orthogonal Training (see (12)). Threshold is the same for each layer of the neural network.

| Dataset | Task 1 | Task 2 | Task 3 | Task 4 | Task 5 | Task 6 | Task 7 | Task 8 | Task 9 | Task 10 |
|---|---|---|---|---|---|---|---|---|---|---|
| PMNIST (iid) | 0.94 | 0.94 | 0.94 | 0.94 | 0.94 | 0.94 | 0.94 | 0.94 | 0.94 | 0.94 |
| PMNIST (non-iid) | 0.96 | 0.96 | 0.96 | 0.96 | 0.96 | 0.96 | 0.96 | 0.96 | 0.96 | 0.96 |
| 5-Datasets | 0.95 | 0.951 | 0.952 | 0.953 | 0.954 | - | - | - | - | - |
| CIFAR100 | 0.87 | 0.871 | 0.872 | 0.873 | 0.874 | 0.875 | 0.876 | 0.877 | 0.878 | 0.879 |
| Mini-Imagenet | 0.90 | 0.901 | 0.902 | 0.903 | 0.904 | 0.905 | 0.906 | 0.907 | 0.908 | 0.909 |

| | Task 11 | Task 12 | Task 13 | Task 14 | Task 15 | Task 16 | Task 17 | Task 18 | Task 19 | Task 20 |
|---|---|---|---|---|---|---|---|---|---|---|
| Mini-Imagenet | 0.91 | 0.911 | 0.912 | 0.913 | 0.914 | 0.915 | 0.916 | 0.917 | 0.918 | 0.919 |

## F.5 GPU Devices

We measured local epoch training time and GPSE time (in Table 4) for computation in NVIDIA Quadro RTX 5000 GPU. We run our experiments in parallel on 8 NVIDIA Quadro RTX 5000 GPUs.

# G Additional Results

## G.1 Effect of Task Heterogeneity

We analyze the effect of task heterogeneity across tasks for each client. In other words, a client does not necessarily see all the tasks. In our experiments, each client see 60% of the tasks on average. In global manner, we follow continual learning literature that is to separate tasks clearly (i.e. two different tasks do not arrive to clients simultaneously). Number of rounds and hyperparameters are set the same as before. However, to simulate the task heterogeneity, we increase the total number of clients so that 60% of that is equal to the number of clients we set for the main experiments for each dataset. By this way, the number of clients that have data for each task is the same as before. We provide the results for task heterogeneity in Table 10.

Table 10: Performance results of different methods on Permute MNIST, Split CIFAR-100, 5-Datasets and mini-Imagenet under task-heterogeneity setting. ACC (higher is better) stands for average classification accuracy of each task at the end of whole CFL rounds, while FGT (lower is better) denotes average forgetting as in (13).

| | Method | PMNIST | | CIFAR100 | | 5-Datasets | | Mini-Imagenet | |
|---|---|---|---|---|---|---|---|---|---|
| | | ACC(%) | FGT(%) | ACC(%) | FGT(%) | ACC(%) | FGT(%) | ACC(%) | FGT(%) |
| IID | FL | 86.93 | 7.15 | 63.89 | 13.51 | 77.86 | 13.62 | 48.43 | 35.34 |
| | EWC+FL | 87.92 | 7.92 | 63.13 | 14.20 | 78.38 | 13.10 | 47.73 | 35.92 |
| | ER+FL | 89.97 | 5.60 | 65.68 | 11.84 | 79.00 | 12.23 | 55.80 | 27.78 |
| | RGO+FL | 89.27 | 6.13 | 62.84 | 16.24 | **85.29** | 2.57 | 50.89 | 31.98 |
| | GLFC+FL | 88.17 | 9.86 | 65.17 | 12.44 | 81.20 | 10.53 | 58.17 | 25.32 |
| | FOT(Ours) | **90.18** | **1.60** | **71.68** | **1.13** | 85.06 | **0.88** | **68.95** | **0.04** |
| non-IID | FL | 79.66 | 9.81 | 63.41 | 9.63 | 66.24 | 24.51 | 43.49 | 31.72 |
| | EWC+FL | 81.56 | 10.49 | 62.41 | 10.14 | 63.87 | 27.90 | 41.49 | 33.69 |
| | ER+FL | 84.71 | 7.11 | 59.43 | 14.07 | 67.14 | 23.47 | 46.32 | 27.36 |
| | RGO+FL | 76.68 | 18.31 | 45.54 | 26.18 | **79.90** | 5.11 | 42.95 | 31.22 |
| | GLFC+FL | **85.36** | 9.85 | 60.74 | 14.20 | 77.65 | 11.42 | 49.58 | 24.93 |
| | FOT(Ours) | **85.61** | **2.16** | **66.03** | **0.93** | 79.37 | **1.99** | **62.10** | **0.19** |

## G.2 Detailed Results

In this section, we provide more detailed results on Split CIFAR-100 and Split mini-Imagenet for each method. In Figure 4, we provide curve of average accuracy measured at the end of each task. For both datasets, FOT outperforms other methods throughout the whole training process. This shows that our method has better performance independent of the number of tasks.

In Figure 5, we provide curves of first task accuracy measured at the end of each task for each method. We can deduce that FOT protects first-task accuracy only with a negligible accuracy drop. The performance of other methods on the first task degrades significantly when new tasks arrive.

In Table 11, we provide the results for Single Task. Single Task stands for the training of each task separately then calculating the average accuracy of each task, i.e. there is different model for each task.

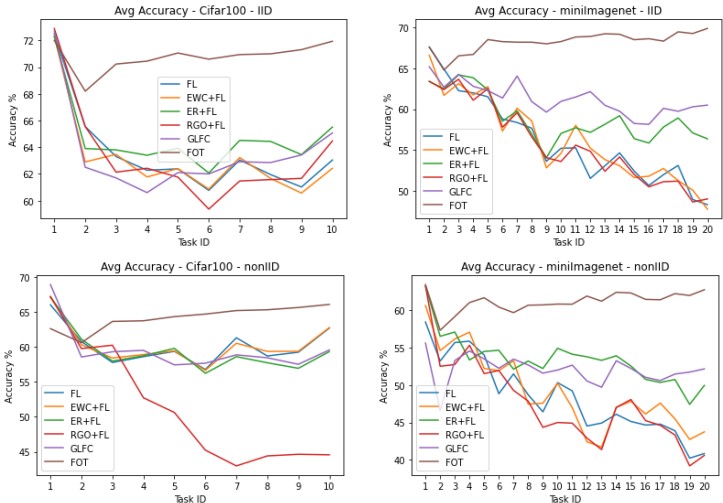

Figure 4: Average accuracies of the model at the end of each task for Split-Cifar100 and Split-Mini-Imagenet Datasets.

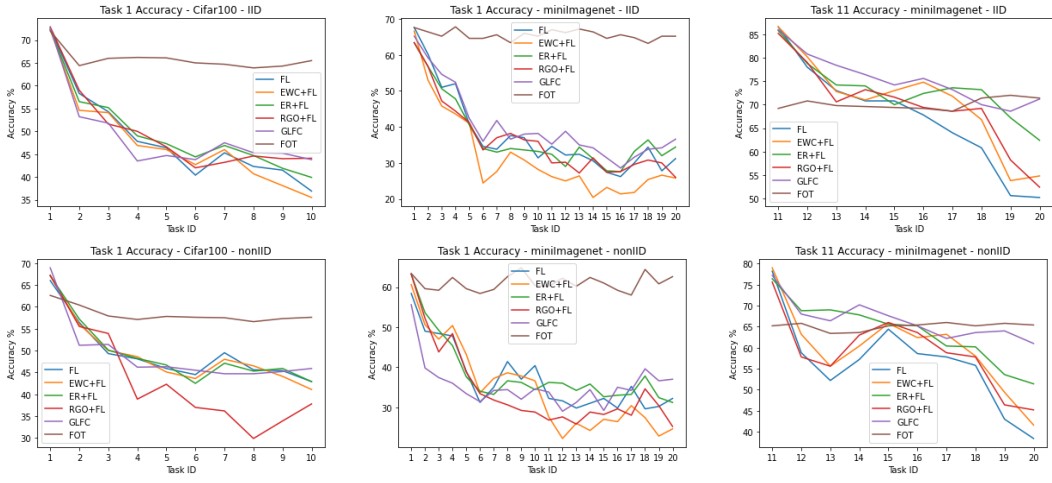

Figure 5: Task 1 and Task 11 accuracies of the model at the end of each tasks for Split-Cifar100 and Split Mini-Imagenet Datasets.

Table 11: Accuracy (%) results of Single Task training on various benchmarks. Please note that FGT is not applicable.

| Method | PMNIST | CIFAR100 | 5-Datasets | Mini-Imagenet |
|---|---|---|---|---|
| Single Task (IID) | 91.10 | 67.83 | 87.74 | 67.60 |
| Single Task (non-IID) | 86.42 | 61.93 | 83.86 | 58.71 |

## G.3 EFFECT OF THRESHOLD IN FOT

We analyze the effect of threshold value in FOT algorithm (12) experimentally. For Split Cifar-100 dataset and under non-IID data distribution among clients, we alter the threshold value of FOT algorithm ranging from 0.80 to 0.96. We provide accuracy curves of all tasks in Figure 6. In Table 12, we provide the numerical results of final accuracy for each task as well as final average accuracy and average forgetting as defined in (13). Also, we demonstrate how the activation space is utilized for all the layers of the neural network (Figure 7).

Table 12: Accuracy of each task at the end of whole training for different threshold values of FOT on Split-Cifar100 benchmark and under non-IID data distribution. ACC and FGT are as defined in (13).

| Threshold | Task1 (%) | Task 2 (%) | Task 3 (%) | Task 4 (%) | Task 5 (%) | Task 6 (%) | Task 7 (%) | Task 8 (%) | Task 9 (%) | Task 10 (%) | ACC (%) | FGT (%) |
|---|---|---|---|---|---|---|---|---|---|---|---|---|
| 0.80 | 51.9 | 57.0 | 70.9 | 66.2 | 69.4 | 70.2 | 72.9 | 70.0 | 72.6 | 76.2 | 67.7 | 2.2 |
| 0.85 | 53.8 | 59.2 | 72.2 | 66.2 | 68.5 | 67.9 | 69.7 | 67.8 | 69.1 | 74.2 | 66.9 | 1.3 |
| 0.87 | 53.8 | 60.5 | 71.8 | 65.0 | 68.9 | 65.9 | 69.3 | 68.0 | 68.5 | 71.5 | 66.3 | 1.1 |
| 0.90 | 57.1 | 61.5 | 70.6 | 64.1 | 66.6 | 64.7 | 65.7 | 65.1 | 67.2 | 69.2 | 65.2 | 0.6 |
| 0.93 | 61.0 | 62.7 | 67.0 | 59.3 | 65.1 | 61.4 | 63.4 | 62.5 | 64.1 | 66.4 | 63.3 | 0.1 |
| 0.96 | 62.4 | 61.6 | 64.2 | 57.9 | 61.7 | 59.3 | 61.2 | 61.0 | 60.6 | 64.5 | 61.4 | -0.1 |

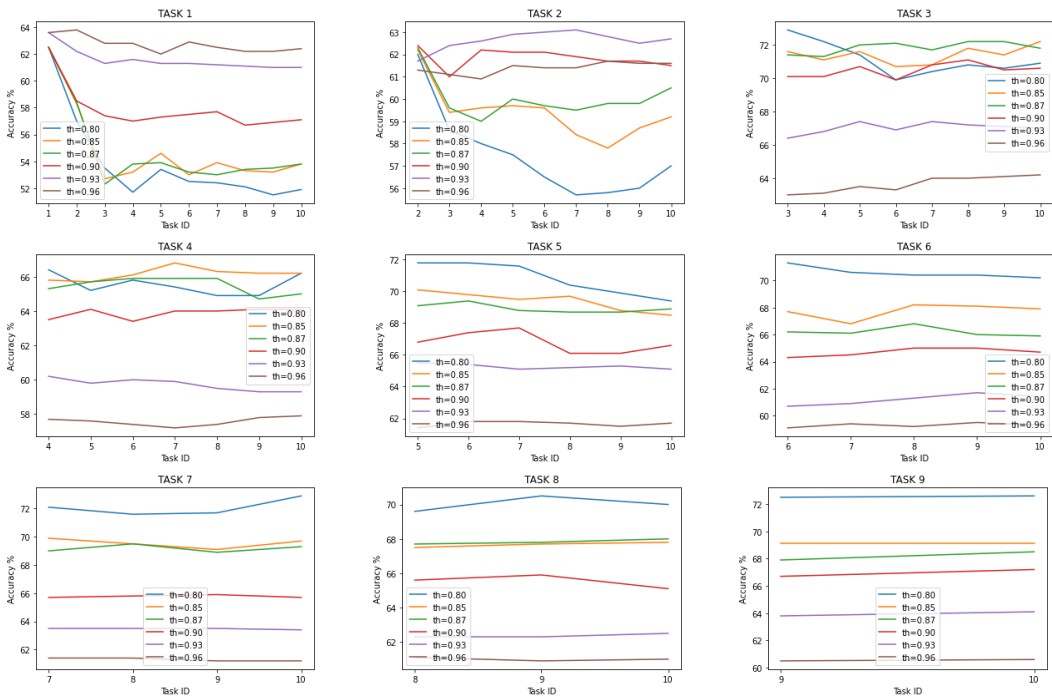

Figure 6: Accuracy of each task measured at the end of training of each task for different threshold values of FOT on Split-Cifar100 benchmark and under non-iid data distribution.

## G.4 COMMUNICATION COST COMPARISON

As we claim in the main paper, the communication cost of FOT is negligible. This is because the only additional cost of FOT is GPSE round, which only happens at the end of each task. Besides, the communicated objects in that round are $\{A_\ell\}_{\ell=1}^{L}$ and $\{O_\ell\}_{\ell=1}^{L}$ which have smaller size than the model itself as we can see from Table 5. This additional cost of FOT per client is fixed. It does not scale with the number of data points or the number of tasks. On the other hand, recent Continual FL work Qi et al. (2023) adds additional communication cost per round. The server and

clients has to communicate an additional generative model after the first task. This increases the total communication cost more than 100%. In Figure 8 and Table 13, we provide the total communication cost of one client at the end of each task for Cifar100 and Mini-Imagenet datasets. At the first task, the communication cost is equal for all methods because they all apply FedAVG. After the first task, the communication cost of FOT and FedAVG is almost equal but cost of FedCIL increases more than others because of the generator model communication.

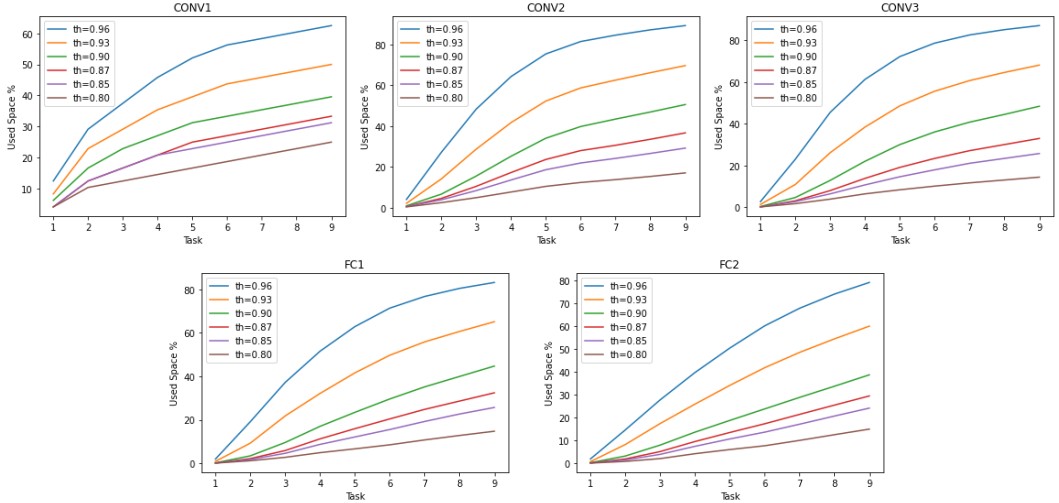

Figure 7: Percentage of used space at the end of each task on Split-Cifar100 (non-IID data distribution) for different threshold levels of FOT.

Table 13: Cumulative communication cost comparison of FedAvg, FOT and FedCIL measured at the end of each task on Split-Cifar100 and Split-mini-Imagenet.

| | | Task1 | Task 2 | Task 3 | Task 4 | Task 5 | Task 6 | Task 7 | Task 8 | Task 9 | Task 10 |
|---|---|---|---|---|---|---|---|---|---|---|---|
| Cifar100 | FedAvg | 2840.0 | 5680.0 | 8520.0 | 11360.0 | 14200.0 | 17040.0 | 19880.0 | 22720.0 | 25560.0 | 28400.0 |
| | FOT | 2840.0 | 5680.3 | 8520.6 | 11360.9 | 14201.2 | 17041.4 | 19881.7 | 22722.0 | 25562.3 | 28402.6 |
| | FedCIL | 2840.0 | 8880.0 | 14920.0 | 20960.0 | 27000.0 | 33040.0 | 39080.0 | 45120.0 | 51160.0 | 57200.0 |
| mini-Imagenet | FedAvg | 1280.0 | 2560.0 | 3840.0 | 5120.0 | 6400.0 | 7680.0 | 8960.0 | 10240.0 | 11520.0 | 12800.0 |
| | FOT | 1280.0 | 2560.7 | 3841.4 | 5122.2 | 6402.9 | 7683.6 | 8964.3 | 10245.0 | 11525.8 | 12806.5 |
| | FedCIL | 1280.0 | 4020.0 | 6760.0 | 9500.0 | 12240.0 | 14980.0 | 17720.0 | 20460.0 | 23200.0 | 25940.0 |
| | | Task11 | Task 12 | Task 13 | Task 14 | Task 15 | Task 16 | Task 17 | Task 18 | Task 19 | Task 20 |
| | FedAvg | 14080.0 | 15360.0 | 16640.0 | 17920.0 | 19200.0 | 20480.0 | 21760.0 | 23040.0 | 24320.0 | 25600.0 |
| | FOT | 14087.2 | 15367.9 | 16648.6 | 17929.4 | 19210.1 | 20490.8 | 21771.5 | 23052.2 | 24333.0 | 25613.7 |
| | FedCIL | 28680.0 | 31420.0 | 34160.0 | 36900.0 | 39640.0 | 42380.0 | 45120.0 | 47860.0 | 50600.0 | 53340.0 |

## G.5 CONVERGENCE ANALYSIS OF FOT

In this section, we provide the convergence curves of FOT and FedAVG for different tasks in Figure 9. As the reader could approve, the accuracy-round curve of FOT and FedAVG exhibits similar convergence behaviour in a task training. One difference is that as we do a convex restriction on the model updates in FOT, the convergence point of FOT is less optimal compared to FedAvg. Therefore, the accuracy of FOT is less than FedAVG at the end of the task. However, when the model moves to the next tasks, FOT achieves to keep that accuracy but FedAVG cannot. This is the reason why FOT have better average accuracy and almost 0 forgetting.

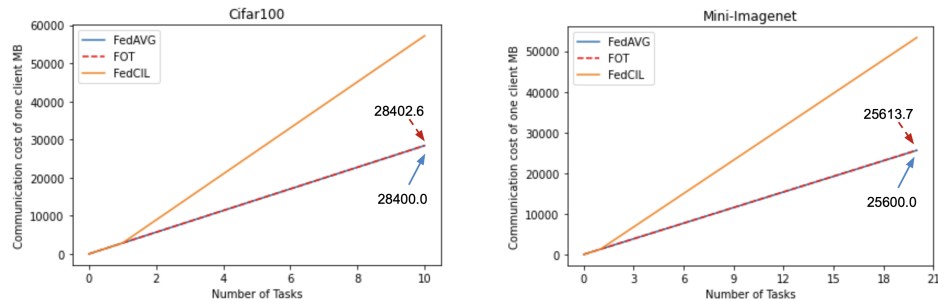

Figure 8: Communication Cost Comparison of FedAvg, FOT and FedCIL

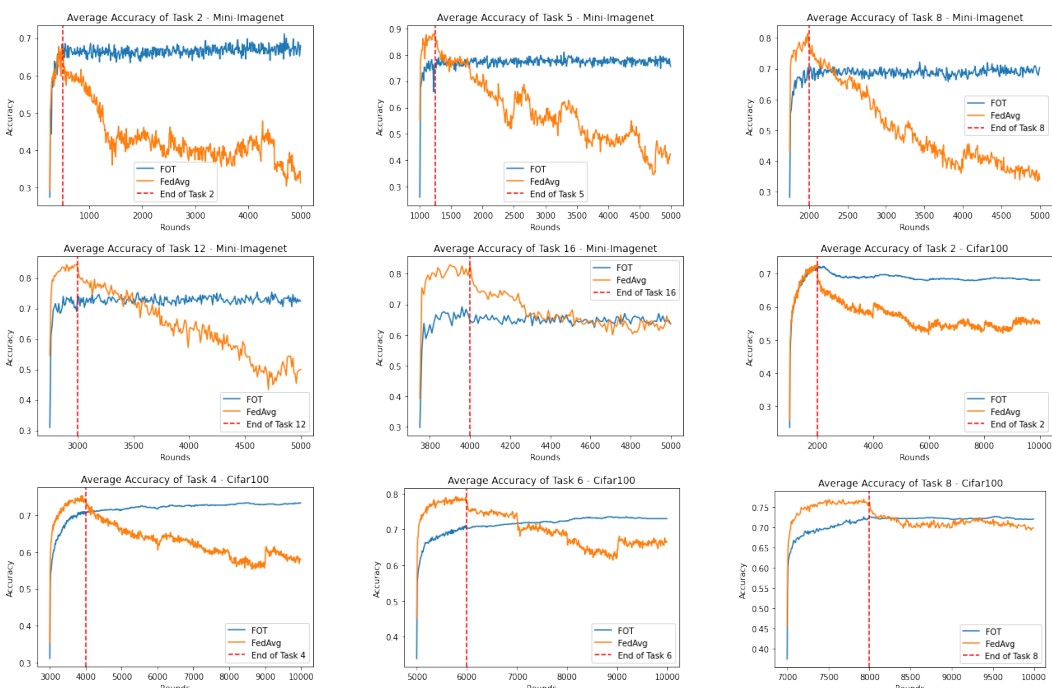

Figure 9: Accuracy - Round Curves of FOT and FedAvg for various tasks in Mini-Imagenet and Cifar100.

## G.6 APPLICATION OF FOT TO SUBSET OF LAYERS

FOT can be applied some layers of the model rather than the whole layers. However, if we exclude some of the layers from FOT, we should choose whether they will be frozen in the later tasks or continue to be trained. If we freeze these layers, FOT still works well as we show in section 5.2.2. In that experiment, we freeze the first layer and exclude it from FOT and the result is still as good as the original FOT. However, if we choose to continue training the excluded layers in the later tasks, FOT might not work well. The intuition behind this is that, when we fine-tune the excluded layers, these layers' activations in response to inputs from earlier tasks could change drastically and this change can propagate to the activation response in later layers in the model.

To empirically show the affect of this, we use FOT on only a subset of layers. In the first experiment, we apply FOT on the last half of the layers and fine-tune the early half of the layers. In the second experiment, we still exclude the first half layers from FOT but we freeze them after the first task. We also do the same experiment where we apply FOT only on the first half of the layers and fine-tune/freeze the second half. The experiments are done in mini-Imagenet with IID distribution and the same hyperparameters as the main experiments are used. Results are shown in Table 14.

When we do not freeze the excluded layers, the average accuracy is low which approves our hypothesis. If we freeze these layers, we still get comparable results but the average accuracy is lower. This is because when we freeze some layers, we restrict the learning of subsequent tasks which leads to learning those tasks worse and lower average accuracy.

Table 14: Performance of applying FOT to the subset of layers on Mini-Imagenet. (A)-(B) denotes that method (A) is applied to the first half of the model and method (B) is applied to the second half.

|  | FOT-FOT | FOT-frozen | FOT-finetune | frozen-FOT | finetune-FOT |
|---|---|---|---|---|---|
| ACC(%) | 69.07 | 67.48 | 49.79 | 67.58 | 47.31 |
| FGT(%) | 0.19 | 0.14 | 28.15 | 0.12 | 33.41 |

## H  ADDITIVE GAUSSIAN MECHANISM

Instead of relying on the privacy guarantees coming from JL transform (Blocki et al., 2012), we formalize the privacy guarantees by applying the Gaussian mechanism (Dwork, 2006) at the clients and use Secure Aggregation to get central approximate differential privacy guarantees at the server. In particular, before sending the JL transformed matrices as in (11), we first clip the transformed activates and then add Gaussian noise as shown in 14 below

$$\mathbf{A}_{t,i}^{\ell} \leftarrow \text{CLIP} \left( \sum_{j=1}^{n_{t,i}} \mathbf{x}_{t,i}^{j,\ell*} \mathbf{g}_j^{\ell T} c \right) + \mathbf{G}_j^{\ell}, \tag{14}$$

where $\text{CLIP}(X|L) = \frac{X}{\|X\|} \min(\|X\|, L)$ (typically we use $L = 1$) and $\mathbf{G}_j^{\ell}$ is sampled from $\mathcal{N}(\mathbf{0}, \sigma^2 \mathbf{I})$. If we choose $\sigma^2 > \frac{\log(1.25/\delta)}{\epsilon^2 C}$, then using standard arguments we can achieve $(\epsilon, \delta)$-central differential privacy, where $C$ is the number of clients in the federated learning system.

