# OpenReview forum: "Federated Orthogonal Training: Mitigating Global Catastrophic Forgetting in Continual Federated Learning"
_ICLR.cc/2024/Conference — ICLR 2024 poster_

### Official Review · Reviewer_VJN9 · 2023-10-22

**Soundness:** 3 good
**Presentation:** 3 good
**Contribution:** 2 fair
**Rating:** 5
**Confidence:** 3

**Summary:**

This paper discusses the concept of Continual Federated Learning (CFL), a real-world scenario where new tasks emerge over time in a decentralized data setting. In CFL, the main challenge is Global Catastrophic Forgetting, where the performance of the global model on old tasks deteriorates as it is trained on new tasks. While previous works have attempted to address this problem, they often rely on unrealistic assumptions about past data availability or violate privacy principles. To overcome these limitations, the authors propose a novel method called Federated Orthogonal Training (FOT). FOT works by extracting the global input subspace of each layer for old tasks and modifying the aggregated updates of new tasks in a way that ensures they are orthogonal to the global principal subspace of old tasks for each layer. This reduces interference between tasks, which is the main cause of forgetting. Empirical evidence shows that FOT outperforms existing continual learning methods in the CFL setting, achieving better average accuracy and lower forgetting while incurring minimal computation and communication costs.

**Strengths:**

1. The paper is well-written.
2. The authors propose a CFL framework named Federated Orthogonal Training (FOT) to address the Global Catastrophic Forgetting problem.
3. Within FOT, they introduce a novel aggregation method, named FedProject, which guarantees the orthogonality in a global manner without privacy leakage and more communication.

**Weaknesses:**

My primary point of concern revolves around the differentiation in technical innovation when compared to related works. More precisely, the paper that introduced the GPM (s Gradient Projection Memory) method for centralized continual learning seems to employ a similar approach to address the issue of forgetting. This similarity is evident in the equations provided – Eq 8 and 9 in the GPM paper and Eq 12 in the current paper.
I acknowledge that this paper introduces an additional layer of aggregation, which generates $A^l$ through secure aggregation in the context of federated learning. However, it's worth noting that secure aggregation inherently lends itself to operations such as summation or averaging. Consequently, I am inclined to question the extent of the technical contribution and novelty offered by this paper. Please correct me if I misunderstand something.

**Questions:**

The author could have a clear statement about the technical novelty and contribution of this paper compared to previous contralized continual learning.

---

> ### Author Response · Authors · 2023-11-14
>
> We thank the reviewer for their feedback and positive comments about technical content.
>
> **Novelty of FOT**
>
> Our paper indeed uses a similar orthogonality idea as in GPM. However, the straightforward adaptation of GPM into federated learning is to make each client apply GPM locally. As shown in Table 1 (see GPM+FL), this does not work well in terms of average accuracy and forgetting. This is because GPM+FL only could extract local subspaces and the updates that are orthogonal to these local subspaces are not necessarily orthogonal to the global subspace after aggregation (See Figure 2). Besides, it adds additional computation cost and storage to the client.
>
> When trying to adopt that orthogonality idea to FL successfully, we need to solve 2 major problems which is our technical novelty in this paper:
>
> 1 - How to make global model updates orthogonal to the previous task layer subspace?
>
> 2 - More importantly, how to get the global principal subspace of layers in a privacy-preserving, communication, and computation cost-friendly way.
>
> In our work we propose 2 methods FedProject and GPSE, addressing problems 1 and 2 respectively. FedProject is a novel aggregation method ensuring that the global model converges to the low-loss regions for all considered tasks and GPSE is proposed to yield the global principal subspace.
>
> *Novelty of GPSE:* Utilization of randomized linear algebra is the crucial part in this algorithm while Secure Aggregation is used as an additional privacy layer. Without randomization, clients would need to directly send the activations to the server. Then, the server would need to concatenate the activations rather than addition/averaging before computing the principal subspace. Recall that the computation of the principal subspace (without the additional random transform in Eq (9) of the manuscript) is a function of the matrix $X$ with each column representing activations for different data points not an aggregated sum of the activations across points. Hence, direct secure aggregation would not be applicable.
> Furthermore, the server would have direct access to the activations (data points itself for the first layer) which violates privacy considerations in FL, and the overall communication overhead for computing the subspace would scale with the number of data points in the FL system.
>
> GPSE solves these issues by utilizing methods in randomized linear algebra in a novel way. By applying random Gaussian transforms to the activations, we ensure that the size of the communicated matrices only grows with the model size rather than the data amount. Also, the required operation to do at the server side becomes linear addition, enabling the use of Secure Aggregation.
> Lastly, the resulting matrix $A^\ell$ on the server side protects the principal column subspace information and does not leak information to the server about the clients, adding a layer of privacy to the system.

---

> > ### Comment · Reviewer_VJN9 · 2023-11-20
> >
> > Thanks for the author's prompt response. I have thoroughly reviewed the response and will discuss it with AC as well as other reviewers in the later phases. For now, I do not have any further questions.

---

> > > ### Author Response · Authors · 2023-11-20
> > >
> > > Thank you for taking the time to review our paper and rebuttal.

---

### Official Review · Reviewer_vBpD · 2023-10-28

**Soundness:** 3 good
**Presentation:** 3 good
**Contribution:** 3 good
**Rating:** 8
**Confidence:** 4

**Summary:**

This Paper works on the continual federated learning problem. The author proposed a framework named Federated Orthogonal Training (FOT) to address the Global Catastrophic Forgetting problem.

**Strengths:**

1. This paper clearly describes the problem setting and states their target -- solving the catastrophic forgetting problem. The authors used illustrations, formulas, and well-written paragraphs to explain the FOT framework step by step clearly.
2. When estimating the global principal, clients need to upload locally extracted principal subspace information to the server. The authors used knowledge from randomized SVD to realize this in a privacy-preserving way.
3. In the paper, the authors discussed several important aspects, especially the privacy, of the algorithm that people worried about.
4. Empirical results showed that the FOT provided significant improvement in average forgetting compared with extensive baselines in various benchmarks

**Weaknesses:**

1. There are still some baselines that are not compared in the paper, for example, the methods in paper [1],[2],[3].
[1] Zhizhong Li and Derek Hoiem. Learning without forgetting. IEEE transactions on pattern analysis and machine intelligence, 40(12):2935–2947, 2017.
[2] Jaehong Yoon, Wonyong Jeong, Giwoong Lee, Eunho Yang, and Sung Ju Hwang. Federated continual learning with weighted inter-client transfer. In International Conference on Machine Learning, pages 12073–12086. PMLR, 2021.
[3] Jie Zhang, Chen Chen, Weiming Zhuang, and Lingjuan Lv. Target: Federated class-continual learning via exemplar-free distillation, 2023.
2. In FOT, the global update was generated by projecting the aggregated updates onto the orthogonal subspace for each layer in the model. Intuitively, this will make the converge slower, which is directly related to potential extra communication costs. The authors did not provide a discussion about this.

**Questions:**

1. On page 6, Theorem 1, the author mentioned that "for sufficiently large n, the principal column space of Y recovers the low-rank column space of A up to rank k with a negligible error.". In my understanding, n corresponds to the number of data in every client. Then what is the number of data in your experiments? For the 100 clients' experiments, is there enough data? If not, how do you explain the effectiveness of the FOT?
2. Authors said that they used the same round number for different methods. How did you decide the round number? Did other methods converge earlier than the round number?
3. According to the description of the FOT, to me, it seems like that FOT would also work well if we only apply this to several layers in the model. Have the authors tried this?

---

> ### Author Response · Authors · 2023-11-14
>
> We thank the reviewer for their feedback and positive comments about technical content. We address your questions below point by point.
>
> **Weakness 1**
>
> *Comparison with Other Baselines:* LwF [1] is a centralized continual learning algorithm proposed in 2017. While determining baselines from centralized CL, our approach involved a thorough examination of both the SOTA methodologies and well-established earlier works. Notably, FedLwF is compared with FedCIL in [A] and shown to perform much worse. Given the inclusion of FedCIL in our baseline, and show that FOT has superior performance in comparison, we have chosen to exclude FedLwF from our analysis.
>
> While categorized under Continual Federated Learning, it's crucial to recognize that FedWeIT [2] addresses a fundamentally distinct problem. It is primarily interested in learning individual local tasks, not learning a global model but utilizing the information of other clients. It employs federated learning not to train a global model but to enhance the continual learning process for individual local tasks by leveraging information from other clients.  Due to the distinct objective of this paper in comparison to ours, we haven’t included it in our baseline.
>
> We mention TARGET [3] in our related work section (Section 2 and Appendix B). Unfortunately, due to its publication date being in close proximity to the submission deadline, we were unable to incorporate it into our baselines.
> Following the reviewer’s advice, we did an initial comparison of TARGET with FOT on mini-Imagenet dataset. At the end of each task, the generator in TARGET is trained on the server side in a data-free way.  Then using that generative model, the server generates synthetic samples (750 samples per task in our experiments) and sends them to the clients. Clients perform knowledge distillation between the current model and the stored old-task model with the synthetic samples to prevent forgetting.
>
> |  | IID |  | nonIID |  |
> |---|---|---|---|---|
> |  | ACC(%) | FGT(%) | ACC(%) | FGT(%) |
> | FedAvg | 50.43 | 33.08 | 41.00 | 32.92 |
> | TARGET | 58.45 | 25.12 | 53.67 | 26.31 |
> |FedCIL | 55.98 | 28.90 | 51.12 | 25.66 |
> | FOT | 69.07 | 0.19 | 62.06 | 0.17 |
>
> Similar to FedCIL, TARGET achieves a better forgetting and accuracy compared to vanilla FedAvg, but is outperformed by our FOT approach. We would like to highlight that these only represent initial comparisons with TARGET and we plan to extend to a more thorough comparison on other datasets in the revised manuscript in Appendix G.7.  Given your suggestions, we have also extended our related work discussion in the manuscript to highlight [1] and [2].
>
> **Weakness 2 & Question 2**
>
> In the pursuit of fairness, we empirically identified the optimal number of rounds for the base case, namely FedAvg, and employed this determined round number consistently across all other methods.
>
> *Convergence of FOT:* Thank you for asking this question. FOT does not exhibit slower convergence; However, it is crucial to highlight that in subsequent tasks, convergence is directed toward the optimal point within a restricted convex set through a subspace projection. While this projection doesn't slow down the convergence speed, it does lead to a slight decrease in that task’s accuracy due to regularization by the target convex set.
> In the updated version of the manuscript (Appendix G.5), we discuss the convergence behaviors of FOT and share the accuracy-round plots. As we can see from these plots, the convergence behavior of FOT is similar to FedAVG. While FOT may exhibit lower accuracy than FedAvg in a specific task, FOT maintains this accuracy in later tasks. On the other hand, the accuracy of other methods tends to decline below FOT's performance as new tasks are introduced. That’s why, at the end of training a sequence of different tasks, the average accuracy of FOT is higher and the forgetting is less than other methods.
>
>
>
>
> **Question 1**
>
> In the server, after aggregating all local $A_{t,i}^\ell$, the server gets global  $A_{t}^\ell$ (see equation 10). This new matrix is equal to the multiplication of global input matrix $X^\ell$ and random Gaussian matrix $G$. Therefore, dimension n from the theorem corresponds to the second dimension of $X^\ell$ which is equal to the total number of samples distributed among clients. The value of n remains constant for the task whether we simulate a task with 100 clients or 50 clients. For the specific quantity of data for each task and dataset, refer to Table 8 in the manuscript, specifically examining the ' # training/task' rows.
>
>
> [A] Daiqing Qi, Handong Zhao, and Sheng Li. Better generative replay for continual federated learning, ICLR, 2023.

---

> > ### Author Response · Authors · 2023-11-14
> >
> > **Question 3**
> >
> > Thank you for asking this question. We could select some layers and apply FOT only on those layers. If we exclude some of the layers from FOT, we should choose whether they will be frozen in the later tasks or continue to be trained. If we freeze these layers, FOT still works well as we show in section 5.2.2. In that experiment, we freeze the first layer and exclude it from FOT and the result is still as good as the original FOT. However, if we choose to continue training the excluded layers in the later tasks, FOT might not work well. The intuition behind this is that, when we fine-tune the excluded layers, these layers’ activations in response to inputs from earlier tasks could change drastically and this change can propagate to the activation response in later layers in the model.
> >
> > Following the reviewer’s advice, we use FOT on only a subset of layers. In the first experiment, we apply FOT on the last half of the layers and fine-tune the early half of the layers. In the second experiment, we still exclude the first half layers from FOT but we freeze them after the first task.  We also do the same experiment where we apply FOT only on the first half of the layers and fine-tune/freeze the second half. The experiments are done in mini-imagenet and the results are shared below. (A)-(B) denotes that method (A) is applied to the first half of the model and method (B) is applied to the second half.
> >
> > |  | FOT-FOT  | FOT-frozen  | FOT-finetune  | frozen-FOT  | finetune-FOT |
> > |---|---|---|---|---|---|
> > | ACC(%) | 69.07 | 67.48 | 49.79 | 67.58 | 47.31 |
> > | FGT(%) |  0.19 |  0.14 | 28.15 |  0.12 | 33.41 |
> >
> > When we do not freeze the excluded layers, the average accuracy is low which is in-line with our hypothesis. If we freeze these layers, we still get comparable results but the average accuracy is lower. This is because when we freeze some layers, we restrict the learning of representations for subsequent tasks which degrades the performance when optimizing for these later tasks.
> >
> > We included the additional experiments described above in Appendix G.6 of the revised manuscript.

---

> ### Comment · Reviewer_vBpD · 2023-11-22
> **Thanks for the response**
>
> The authors have addressed all my concerns and I have raised my score.

---

> > ### Author Response · Authors · 2023-11-22
> > **Expressing Thanks for Reviewer's Valuable Input**
> >
> > We extend our thanks to the reviewer for their valuable input in improving our paper and acknowledging its value. Your support is greatly appreciated.

---

### Official Review · Reviewer_Zv1b · 2023-10-31

**Soundness:** 3 good
**Presentation:** 3 good
**Contribution:** 3 good
**Rating:** 6
**Confidence:** 3

**Summary:**

To address the catastrophic forgetting problem in the context of federated continual learning, the paper introduces a novel method that leverages orthogonalization of tasks to mitigate global forgetting in the course of continuous learning. This approach effectively reduces interference between distinct tasks, as demonstrated empirically. FOT exhibits superior performance compared to existing methods, manifesting improvements in accuracy and reduction in forgetting rates, all while incurring minimal additional computational and communication costs.

**Strengths:**

The study presents a CFL framework, Federated Orthogonal Training (FOT), which addresses Global Catastrophic Forgetting by modifying global updates for new tasks to reduce interference with previous tasks. FOT also ensures client privacy, eliminates the need for client-side storage, and outperforms other methods in cross-device settings, even though they have additional computation and storage requirements.

**Weaknesses:**

Communication overhead represents a significant weakness in the methodology presented in the paper. However, the paper's analysis is somewhat superficial, lacking a comparison with baseline experiments. The argument regarding the minimized communication overhead is not sufficiently elaborated upon.

Similarly, Remark 1 highlights that the convergence analysis in the paper is relatively straightforward and lacks comprehensive theoretical underpinnings.

**Questions:**

I have a question of how to understand the correctness of the orthogonal process in this paper (intuitively)?

---

> ### Author Response · Authors · 2023-11-14
>
> We thank the reviewer for their feedback and positive comments about technical content. We address your questions below point by point.
>
> **Communication Overhead:**
> We would like to emphasize that the additional communication cost incurred by FOT is negligible. First, it is important to note that this additional cost is associated only with the GPSE step, and this step consists of just one round, conducted once at the end of each task. In this round, the server transmits $\\{ O_\ell \\}^L_{\ell=1}$ to the clients, and the clients send only $\\{ A_\ell \\}^L_{\ell=1}$ to the server. Importantly, the communication cost of GPSE remains independent of the amount of data and is scaled only with the model size. Furthermore, while the communication complexity aligns with the model size, the actual size is observed to be smaller than the model itself as shown in Table 5.
>
> As a numerical example from our experiments, if we train Alex-net on one Cifar100 task for 1000 federated learning rounds by communicating the model, then the FOT cost and FedAVG incur exactly the same cost (2840 MB in total) over the 1000 model training rounds because FOT only communicates the model between clients and the server (as in FedAvg).  At the end of 1000 rounds, GPSE uses a communication load less than or equal to only a single round which is much smaller than rounds used for model aggregation because GPSE communicates $\\{ O_\ell \\}^L_{\ell=1}$  and $\\{ A_\ell \\}^L_{\ell=1}$ and their total size (0.29 mb - See table 5) is less than model size (1.42 mb - See table 5 ). As a result, the additional communication cost is less than 0.1% of the total training cost for a single task. That’s why we claim the additional communication overhead is minimal.
>
> In contrast, recent federated continual learning work -  FedCIL [1] has a huge overhead in terms of communication. FedCIL communicates a generative model with the original classifier in each training round (compared to FOT which only does additional communication only once per task). Given the generative model is bigger (1.60 mb) than the classifier, it increases the communication cost by more than 100%.
>
> We provide the total communication cost (in MB) of one client at the end of each Cifar100 task in the below table. We give a more extensive description of the additional communication cost in Appendix G.4 in the updated version of the paper.
>
> | Task | 1 | 2 | 3 | 4 | 5 | 6 | 7 | 8 | 9 | 10 |
> |---|---|---|---|---|---|---|---|---|---|---|
> | FedAvg | 2840.0 | 5680.0 | 8520.0 | 11360.0 | 14200.0 | 17040.0 | 19880.0 | 22720.0 | 25560.0 | 28400.0 |
> | FOT | 2840.0 | 5680.3 | 8520.6 | 11360.9 | 14201.2 | 17041.4 | 19881.7 | 22722.0 | 25562.3 | 28402.6 |
> | FedCIL | 2840.0 | 8880.0 | 14920.0 | 20960.0 | 27000.0 | 33040.0 | 39080.0 | 45120.0 | 51160.0 | 57200.0 |
>
> **Convergence Analysis:** The paper presents an approach for enabling continual learning in a federated learning setting. We rely on an orthogonalization approach, however, we take careful consideration in allowing this without violating the privacy requirements and the convergence guarantees in FL. Our proposed approach allows us to achieve protection against forgetting using projection into a convex set which allows us to rely on convergence guarantees in the literature for ProximalSGD. We would argue that this simplicity is a strength of the proposed method as it allows us to guarantee theoretical convergence with privacy guarantees and little communication overhead in addition to observed empirical benefits.
>
> We would also like to point out that the central theoretical challenge in the considered problem and our proposed approach is enabling the learning of principal subspaces and the application of orthogonal projection in a distributed privacy-preserving setting.
>
>
> [1] Daiqing Qi, Handong Zhao, and Sheng Li. Better generative replay for continual federated learning, ICLR, 2023.

---

> > ### Author Response · Authors · 2023-11-14
> >
> > **Intuition of Orthogonal Process:** To understand why the orthogonal process works intuitively, let's say we train a model on a given task t. When we move to the next task t+1, the data of task t will disappear. That’s why, while training task t+1, the new model updates will disrupt what the model learns from task t. With an orthogonal process, we restrict the update of the model such that the disruption to the layer outputs (activations) for task t will be very small.
> >
> > We examine the model layer by layer and we will apply the orthogonal process to each layer. Let $X_t$ be the input of a layer at task t and $W$ be the optimal weight of that layer at the end of task t. When we move to the next task, $W$ will be updated as we do training. So, the model’s new response to previous task input is $(W + \Delta W)X_t$. Our goal is to make the change in the layer response to task t input as minimal as possible (ideally zero). If the response of the new model is only minimally changed with respect to the old model at all layers, then the final model output remains the same (or very close), which means no (minimal) forgetting.
> > An easy way to visualize this is if we ideally make $\Delta W X_t = 0$, then $(W + \Delta W)X_t = W X_t$, meaning training of t+1 didn’t affect task t.
> > Therefore, we aim to make  $\Delta W X_t = 0$ ideally or more practically, we want to make  $|| \Delta W X_t|| \approx  0$.
> > To achieve it, we should make the column space of $X_t$ orthogonal to the row space of $\Delta W$, which is the crux of the orthogonal process presented in the paper.
> >
> > To realize this objective in a federated learning setup, we need 2 things:
> >
> > 1- Extracting  principal column subspace of $X_t$ correctly
> >
> > 2-Ensuring updates are orthogonal to that subspace.
> >
> > GPSE achieves 1 in a privacy-preserving, communication, and computation-friendly way and FedProject achieves 2 in the server side.

---

> > ### Comment · Reviewer_Zv1b · 2023-11-23
> > **Thanks for the response!**
> >
> > I've read all responses, and think this is a good paper for accept.

---

> > > ### Author Response · Authors · 2023-11-23
> > > **Thanks for the feedback**
> > >
> > > We thank the reviewer for the feedback and positive comments!

---

### Author Response · Authors · 2023-11-18
**General Comment to All Reviewers**

Dear Reviewers,

We hope that we have clarified all the questions in the initial review. In addition to our clarifications, we have revised the paper and also added numerical experiments to the Appendix to compare with the highlighted methods and proposals in the review, as well as further discussion on the communication cost and convergence behavior of our FOT method  (highlighted in blue in the new version). We had not seen any further comments by the reviewers. If you have any further questions, please let us know. We would appreciate it if you could consider increasing your evaluation scores, given our responses.

---

### Meta-Review · Area_Chair_tWnT · 2023-12-11

**Metareview:**

The reviewers appreciated the novelty of the method, the clever use of private SVD, and the fact that the proposed scheme outperforms competitors. Some very recent competitors were missing from experiments, but the reviewers added them at rebuttal time, these results would be great to add to the main body of the paper. It would be great also if some discussion on communication costs made it to the supplement at least, as well as the discussion on the intuition behind the approach. Finally, the discussion on the novelty over related work should also make it into the paper.

**Justification For Why Not Higher Score:**

Though ok with acceptance, reviewers remained lukewarm.

**Justification For Why Not Lower Score:**

Everyone agreed on acceptane.

---

### Decision · Program_Chairs · 2024-01-16

Accept (poster)